# γδ T cells control murine skin inflammation and subcutaneous adipose wasting during chronic *Trypanosoma brucei* infection

Juan F. Quintana [1,2,8,10] ✉, Matthew C. Sinton [1,2,9,10], Praveena Chandrasegaran [1,2], Agatha Nabilla Lestari[1,2], Rhiannon Heslop [1,2], Bachar Cheaib[1,2,3], John Ogunsola [1,2], Dieudonne Mumba Ngoyi [4], Nono-Raymond Kuispond Swar[1,4], Anneli Cooper [1,2], Neil A. Mabbott [5], Seth B. Coffelt [6,7] & Annette MacLeod [1,2] ✉

African trypanosomes colonise the skin to ensure parasite transmission. However, how the skin responds to trypanosome infection remains unresolved. Here, we investigate the local immune response of the skin in a murine model of infection using spatial and single cell transcriptomics. We detect expansion of dermal IL-17A-producing Vγ6+ cells during infection, which occurs in the subcutaneous adipose tissue. In silico cell-cell communication analysis suggests that subcutaneous interstitial preadipocytes trigger T cell activation via *Cd40* and *Tnfsf18* signalling, amongst others. In vivo, we observe that female mice deficient for IL-17A-producing Vγ6+ cells show extensive inflammation and limit subcutaneous adipose tissue wasting, independently of parasite burden. Based on these observations, we propose that subcutaneous adipocytes and Vγ6+ cells act in concert to limit skin inflammation and adipose tissue wasting. These studies provide new insights into the role of γδ T cell and subcutaneous adipocytes as homeostatic regulators of skin immunity during chronic infection.

The skin represents the ultimate barrier, offering physical and mechanical protection against insults including infections. The skin is a complex organ encompassing several tissue layers, including the dermis and epidermis, which are capable of extensive remodelling and resistance to mechanical and chemical stimuli. Moreover, the skin hosts a complex stromal microenvironment containing a myriad of resident and recruited immune cells that actively survey the tissue. Amongst the dermal immune compartment, γδ T cells have emerged as critical regulators of tissue homeostasis[1–4]. For instance, Vγ6+ cells are enriched in mucosal tissues[3] and the dermis[2,4] where they are considered resident cells. These skin resident γδ T cells express a wide range of effector molecules including interleukin 17 A (IL-17A) and IL-17F[5], IL-22[6], and the epidermal growth factor (EGF) receptor ligand[7], and amphiregulin (AREG)[8], which are critical for promoting tissue repair and limiting inflammation following an insult[9–12]. Although γδ T cells have gathered a lot of attention for their role in cancer[13–16], and

[1]Wellcome Centre for Integrative Parasitology (WCIP), University of Glasgow, Glasgow, UK. [2]School of Biodiversity, One Health, Veterinary Medicine (SBOHVM), College of Medical, Veterinary and Life Sciences, University of Glasgow, Glasgow, UK. [3]Translational Lung Research Center Heidelberg (TLRC), Center for Infectious Diseases, Heidelberg University Hospital, 69120 Heidelberg, Germany. [4]Department of Parasitology, National Institute of Biomedical Research, Kinshasa, Democratic Republic of the Congo. [5]The Roslin Institute and Royal (Dick) School of Veterinary Studies, University of Edinburgh, Edinburgh, UK. [6]School of Cancer Sciences, University of Glasgow, Glasgow, UK. [7]Cancer Research UK Beatson Institute, Glasgow, UK. [8]Present address: Division of Immunology, Immunity to Infection and Respiratory Medicine, Lydia Becker Institute of Immunology and Inflammation. University of Manchester, Manchester, UK. [9]Present address: Division of Cardiovascular Sciences, University of Manchester, Manchester, UK. [10]These authors contributed equally: Juan F. Quintana, Matthew C. Sinton. ✉e-mail: juan.quintana@glasgow.ac.uk; annette.macleod@glasgow.ac.uk

autoimmune disorders affecting the skin such as psoriasis[17] and lupus[18], their function during parasitic colonisation of the skin is still not well understood.

Despite the robust nature of the skin as a barrier, some pathogens manage to circumvent its defences. Of these, trypanosomatids, including the causative agents of leishmaniasis, chagas disease, and sleeping sickness, are known to colonise the skin in a process that is postulated to be critical for infection and parasite transmission[19–21]. Indeed, we and others recently reported the presence of *Trypanosoma brucei* in the skin of mice and humans[21–23]. Such skin infections are often associated with dermatitis and pruritus[22] or can occur in the absence of clinical symptoms or a positive result from gold-standard diagnostic methods[22]. This strongly indicates that mild or asymptomatic skin infections act as important but overlooked parasite reservoirs hampering ongoing eradication efforts. However, to date, the skin response to *T. brucei* infection remains unknown.

Here, we used a combined single cell and spatial transcriptomic approach to generate a spatially resolved single cell atlas of the murine skin during *T. brucei* infection. This combined approach led us to identify that interstitial preadipocytes and Langerhans cells both have central roles in the production of local cytokines and antigenic presentation, taking place largely in the subcutaneous adipose tissue and in proximity to the subcutaneous adipose tissue, respectively. Furthermore, we identified a population of skin IL-17A-producing Vγ6+ cells, located in proximity to adipocytes in the subcutaneous adipose tissue, that display features of TCR engagement and T cell activation upon infection. Cell-cell communication analyses between subcutaneous adipocytes and Vγ6+ cells predicted several potential interactions mediating not only T cell activation via *Tnfsf18*, but also promoting adipocyte metabolism via Vγ6+ cell-derived *Clcf1* and *Areg*. In vivo analyses revealed that Vγ4/6 γδ T cell-deficient mice experience severe skin inflammation with limited subcutaneous adipose tissue wasting compared to infected wild type controls. We conclude that IL-17A-producing Vγ6+ cells are critical mediators of skin immunity against *T. brucei* infection, likely acting both on restraining IFNγ-mediated CD8+ T cell responses in the skin, and promoting subcutaneous adipose tissue wasting, supporting our recently identified role of IL-17 signalling as a mediator of adipose tissue wasting during *T. brucei* infection[24]. More broadly, our results provide a spatially resolved atlas that can help to dissect further immunological pathways triggered in the skin in response to infection.

## Results

### Spatially resolved single cell atlas of the murine skin chronically infected with African trypanosomes

To study the immune responses of the murine skin against African trypanosomes, we conducted a combined single cell and spatial transcriptomic analysis using platforms from 10X Genomics (materials and methods). Skin from female BALB/c mice infected with *T. brucei* GVR35 displayed marked inflammation, as determined by examination of H&E staining (Fig. 1A), compared to uninfected skin. Parasites were detected in extravascular spaces (Fig. 1B) at 21 days post-infection. We chose to explore this time point as the *T. brucei* skin infection is well established, enabling us to explore how the skin adapts to such conditions. Following skin dissociation and scRNAseq analysis, we obtained a total of 56,876 high-quality cells with an average of 1622 genes and 29,651 reads per cell (Fig. 1C, Supplementary Fig. 1, and Supplementary Data 1) from naïve (n = 2) and infected (n = 2) mice. These cells were broadly classified into 16 different clusters (Fig. 1C) based on common expression makers putatively associated with these clusters (Fig. 1D and E). The stromal compartment consisted of eight keratinocyte clusters (46,591 cells; KC 1 to 8), characterised by a high expression level for *Krt14, Krt15, Krt35, and Fabp5*, in addition to one cluster with high expression of *Col1a1, Ly6a, Thy1, and Pdgfra* that we designated as fibroblast-like mesenchymal cells (4274 cells), one *Cd36*+

*Cldn5*+ *Pecam1*+ endothelial cell cluster (792 cells), one *Pparg*+ adipocyte (529 cells), and one *Mlana*+ melanocyte cluster (755 cells) (Fig. 1C and D). A closer examination of differences between experimental conditions allowed us to detect the expansion of individual cell clusters in response to infection. For example, the *Lyz2*+ myeloid and *Cd3g*+ T cell clusters have higher frequencies in the infected skin compared to naïve controls (Fig. 1D). Similarly, we noted the presence of a small erythrocyte cluster exclusively in the infected sample, which may be indicative of infection-induced vascular leakage (Fig. 1D). These results are consistent with the increased frequency of inflammatory cells in infected samples that we determined by histopathological examination.

In the spatial context, the predicted distribution of the various stromal cell populations identified in our single cell atlas was heterogeneous (Fig. 1F, G, and Supplementary Fig. 2A). For instance, the keratinocyte and fibroblast clusters were predicted to be predominantly found in the epidermal layers of the skin biopsies, in particular keratinocytes clusters 2 and 3 (Supplementary Fig. 2B), whereas the melanocytes and adipocytes, and discreet keratinocyte clusters (clusters 4, 7, and 8) were preferentially predicted to be in the dermis and hypodermis in both skin biopsies examined (Fig. 1F and Supplementary Fig. 2B). We also noted that Langerhans and myeloid cells (containing macrophages and dendritic cells) were preferentially found in the epidermis of the skin of naïve animals, around the perifollicular spaces as recently found in human skin[25]. During infection, Langerhans and myeloid cells also localised to the dermal and hypodermal regions, localising with the adipose tissue (Fig. 1F and Supplementary Fig. 2B). The skin immune compartment consisted of *Cd3g*+ *Trdc*+ T cells (1103 cells), *Cd207*+ Langerhans cells (LCs; 854 cells), *Lyz2*+ myeloid cells (1499 cells), and erythrocytes (479 cells) (Fig. 1C and D). The majority of these immune cells were lowly detected in naïve skin but were readily found in the subcutaneous adipose tissue of infected samples (Fig. 1G and Supplementary Fig. 2B). Cells within the myeloid (including Langerhans cells) and Lymphoid compartment (T cells) were identified mostly in the epidermis in the naïve skin section (Fig. 1G and Supplementary Fig. 2B) but were also predicted to occupy niches within the adipose tissue layer in both naïve and infected samples (Fig. 1G and Supplementary Fig. 2B). Together, these data predict a local distribution of immune and stromal cells within the subcutaneous adipose tissue layer in response to *T. brucei* infection.

### Skin preadipocytes and keratinocytes provide inflammatory signals and antigen presentation during chronic *T. brucei* infection

Stromal cells are increasingly recognised as critical for initiating immunological responses in many tissues, including the skin[26]. To capture as much stromal cell resolution as possible, and to better understand how this compartment in the murine skin responds to infection, we reclustered stromal cell clusters (keratinocytes, adipocytes, endothelial cells, melanocytes, and fibroblast-like mesenchymal cells) and reanalysed them separately. After re-clustering, we obtained a total of 13 different stromal clusters, seven of which were bona fide keratinocytes expressing *Krt25, Krt14, and Krt15* (Fig. 2A and B). In addition to these clusters, we also detected *Mlana*+ melanocytes, *Plin2*+ mature adipocytes, and *Cldn5*+ endothelial cells (Fig. 2A and B). Interestingly, the re-clustering analysis led us to identify a clear population of cells expressing bona fide mesenchymal genes including *Cd34, Thy1, Ly6a, Pdgfra, Pdgfrb, and Col1a1* (Fig. 2A and B). We assigned these cells as interstitial preadipocytes (stem-like adipocyte precursor cells) as they also express *Dpp4, Pi16, and Dcn* (Fig. 2A and B), markers previously shown to be expressed in this cell type[27]. These two populations of *Dpp4*+ *Pi16*+ interstitial preadipocytes (IPAs), *Thy1*+ IPA1 and *Cd34*+ *Ly6a*+ IPA2, likely represent cells at different developmental stages within the adipocyte trajectory. Some stromal populations were altered during infection. For instance, we observed a higher frequency

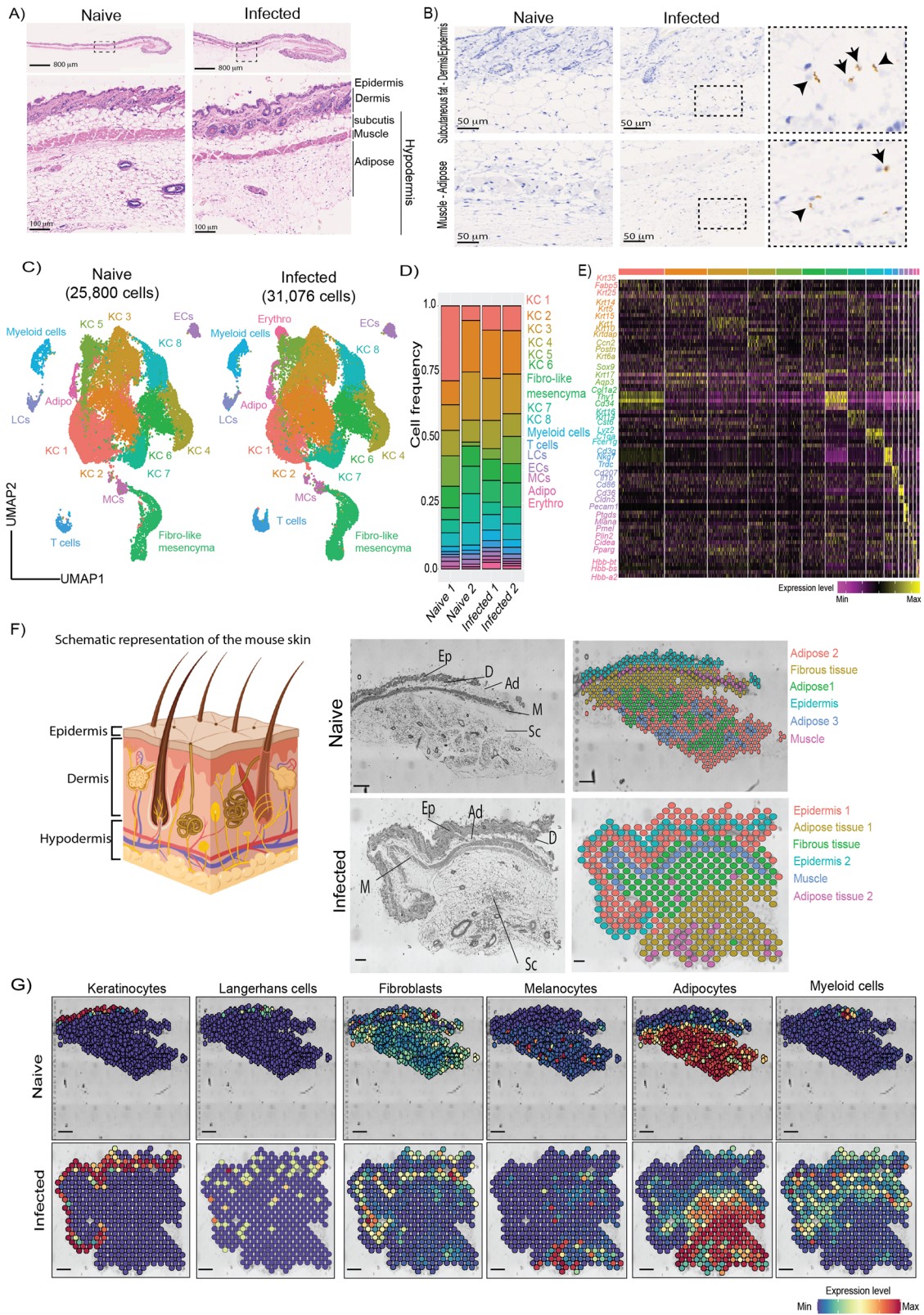

of endothelial cells (ECs; from 6.67% to 8.02% in naïve and infected skin, respectively), IPA1 (from 4.01% to 5.81% % in naïve and infected, respectively), and IPA2 (from 1.98% to 3.81% % in naïve and infected, respectively) in infected compared with naïve skin (Fig. 2C).

Stromal cells are recognised as drivers of inflammatory responses *via* the production of cytokines and chemokines[26], as well as the induction of T cell activation via antigen presentation[28]. To unbiasedly

assess our transcriptomics dataset, to determine whether stromal cells were influencing immune function, we performed module scoring. This was used to determine the global expression levels of cytokines (Fig. 2D) and antigen presentation molecules (Fig. 2E). This analysis revealed that IPA2 (interstitial preadipocyte 2), and to a lesser extent KC2 (keratinocyte 2), were the main producers of cytokines and antigen presentation molecules (Fig. 2D and E). Indeed, almost exclusively

**Fig. 1 | Integrative overview of the murine skin infected with *T. brucei* infection using single cell transcriptomics. A** Representative H&E-stained images from naïve and infected mice from three independent experiments. Scale bar: 800 μm (top) and 100 μm (bottom). **B** Immunohistochemistry against the stumpy-specific marker PAD1 in naïve and infected samples, including insets highlighting the presence of stumpy forms in both the epidermis-subcutaneous adipose tissue and the subcutaneous adipose tissue-adipose tissue. Scale bar: 50 μm. **C** Uniform manifold approximation and projection (UMAP) of 56,876 high-quality cells from both naïve (left panel) and infected (right panel) samples, highlighting the major cell types detected in our dataset, including stromal cells (Keratinocytes, fibroblasts, endothelial cells, and adipocytes) and immune cells (myeloid cells, T cells, Langerhans cells, and macrophages). **D** Frequency plot of the major cell types identified in the murine skin over the course of infection across biological replicates (*n* = 2 replicates/experimental condition). **E** Top 10 marker genes defining all the major cell clusters detected in (**C**). The heatmap is colour-coded based on gene expression intensity. **F** Left panel: schematic representation of the murine skin highlighting the three main skin layers: epidermis, dermis, and hypodermis (encompassing the adipose tissue and muscle). Created with Biorender (Agreement number AT25PE-TUME). Right panel: Tissue sections and spatial UMAP plot of the samples included in the spatial transcriptomics analysis. The major skin compartments are highlighted and include the epidermis (Ep), dermis (D), adipose tissue (Ad), muscle (M), and subcutis (Sc). Scale bar, 100 μm. **G** Integration of single cell and spatial transcriptomics datasets for the major cell types typically found in the skin, including keratinocytes, Langerhans cells, fibroblasts, melanocytes, adipocytes, and myeloid cells using the Seurat package. Scale bar, 100 μm. keratinocytes (KC), fibroblasts (FB), endothelial cells (EC), Macrophages (MC), Langerhans cells (LC), adipocytes (Adipo), erythrocytes (Erythro).

in IPA2, we noted increased expression of *Il6, Il15, Il18bp* and interferon-driven *Cxcl9, Cxcl10*, as well as the class Ib and II major histocompatibility complex *H2-M3* and *H2-DMa*, respectively (Fig. 2F–H). In the spatial context, most of these responses were identified as restricted foci in the dermis and epidermis of naïve mice (Fig. 2I), but significantly localised to the subcutaneous adipose tissue in infected mice (Fig. 2I). Taken together, these results indicate that the IPA2 subcluster, located in the subcutaneous adipose tissue, is a key driver of recruitment and activation of immune cells in the skin in response to *T. brucei* infection.

## Langerhans cells, dendritic cells, and *Cd14*⁺ monocytes contribute to antigen presentation in the murine skin during chronic *T. brucei* infection

In addition to the stroma, resident myeloid cells are also involved in the initial responses to infection. Thus, we next examined transcriptional responses within the myeloid compartment, encompassing *Lyz2*⁺ myeloid and *Cd207*⁺ Langerhans cells (Fig. 1C and D). To gain as much granularity as possible within this compartment, we reclustered the myeloid subset leading to the identification of six subclusters including *Cd14*⁺ monocytes, *Mrc1*⁺ macrophages, two clusters of *Cd207*⁺ Langerhans cells, *Hdc*⁺ *Gata2*⁺ mast cells, and *Clec9a*⁺ *Btla*⁺ *Xcr1*⁺ dendritic cells, which we classed as cDC1s (Fig. 3A and B). Interestingly, in addition to *Cd14*⁺ monocytes, *Mrc1*⁺ macrophages also expressed *Il10*, suggesting that these cells may have anti-inflammatory properties (Fig. 3B). We found that upon infection, there is an expansion of all the myeloid cells, but notably so within the *Mrc1*⁺ macrophage, *Cd14*⁺ monocyte and *Cd207*⁺ Langerhans cell subclusters (Fig. 3A). During infection, the transcripts associated with these subsets, as well as the mast cells and cDC1s, were predicted to be enriched in the subcutaneous adipose tissue (Fig. 3C), mirroring the predicted localisation of other stromal cells driving immune cell recruitment and activation (Fig. 2I). Furthermore, *Cd207*⁺ LCs were predicted to be predominantly enriched in the epidermis/dermis of naïve animals but were also found within the adipose tissue in skin sections from infected animals, coinciding with adipocytes and several populations of progenitor keratinocyte populations (Figs. 1E, 3C, and Supplementary Fig. 2B). The spatial enrichment of LCs in skin biopsies from naïve and infected animals was independently validated using smFISH, which confirms the accumulation of *Cd207*⁺ LCs and *Cd4*⁺ T cells around the perifollicular spaces in the epidermis/dermis in naïve samples, but were also detected in the adipose tissue within the hypodermis during infection (Fig. 3D), consistent with previous studies demonstrating that CD207⁺ LCs display migratory behaviour during infection[29].

Given their role in initiating an adaptive immune response, we next focussed on characterising myeloid cell function by measuring the global expression levels of pro-inflammatory cytokines and molecules associated with antigen presentation, as we did for the stromal compartment, using module scoring. This analysis revealed that the *Cd14*⁺ monocytes and *Mrc1*⁺ macrophages express the highest levels of

cytokines within the myeloid compartment (Fig. 3E). Other immune cells within this cluster, in particular mast cells, are also likely to contribute to the pool of cytokines produced locally. For instance, the mast cell cluster expresses high levels of *Il4, Il13*, and *Csf1* (Supplementary Data 1D), suggesting that these cells engage with Th2 responses to promote tissue repair. In contrast, the antigenic presentation was predominantly driven by Langerhans cells and DCs (Fig. 3F). Taken together, these analyses reveal that myeloid cells located in the subcutaneous adipose tissue drive the expression of pro-inflammatory cytokines and antigen presentation concertedly with stromal cells during infection.

## Both *Cd4*⁺ T cells and Vγ6⁺ cells expand in the skin during chronic *T. brucei* infection

We next examined the T cell compartment in our scRNAseq dataset. After re-clustering, we identified a total of 1043 cells encompassing *Il5*⁺ *Gata3*⁺ ILC2s (138 cells), *Ncr1*⁺ NK cells (172 cells), *Icos*⁺ *Rora*⁺ CD4⁺ T cells (426 cells), and two separate cell clusters of *Trdc*⁺ γδT cells, 1 (155 cells) and 2 (152 cells) (Fig. 4A and B). Both of the γδ T cell clusters express high levels of *Tcrg-C1, Cd163l1*, but low levels of *Cd27* (Fig. 4B), and based on recent literature, this indicates that they are likely to be IL-17-producing Vγ6⁺ cells[30]. Overall, we noted an expansion within the T cell compartment in response to infection. For instance, we noted a 4.2-fold increase in the frequency of *Ncr1*⁺ NK cells (5.2% vs 21.9% in naïve and infected skin, respectively), a 1.35-fold increase in the frequency of ILC2s (9.86% vs 13.37% in naïve and infected skin, respectively) (Fig. 4A). Intriguingly, we noted that the Vγ6⁺ cell cluster 1 was present in the naïve skin but reduced upon infection (40.38% vs 5.09% in naïve and infected skin, respectively) (Fig. 4A). In contrast, there was a 92-fold increase in the frequency of cells within Vγ6⁺ cluster 2 cells in response to infection (0.22% vs 20.88% in naïve and infected skin, respectively) (Fig. 4A). In vivo, we observed a significant expansion of CD27⁻ (IL-17A-producing) γδ T cells and a concomitant reduction in the frequency of CD27⁺ (IFNγ-producing) γδ T cells in skin biopsies from infected BALB/c mice compared to naïve controls (Fig. 4C), following the same trend predicted by the scRNAseq data (Fig. 4A and B), thus validating our in silico prediction. Additionally, we detected a significant expansion in the population of IL-17⁺ Vγ6⁺ cells in the skin of infected mice compared to naïve controls (Supplementary Fig. 3A and B), where Vγ6⁺ cells represent approximately 40% of all the dermal γδ T cells (Supplementary Fig. 3A and B), confirming our in silico predictions.

Based on these findings, we hypothesised that these two γδT cell clusters represent different activation states, whereby γδT cells within cluster 1 represent "resting" cells, and γδT cells within cluster 2 represent "activated" cells. To explore this hypothesis in more detail, we examined the expression level of γδT cell activation and replication markers, namely, *Cd44, Cd69, Nr4a1*, and *Mki67*. We found that, upon infection, the γδT cells within cluster 2 robustly express *Cd44* and *Nr4a1*, and to a lesser extent *Mki67*, suggesting local activation and

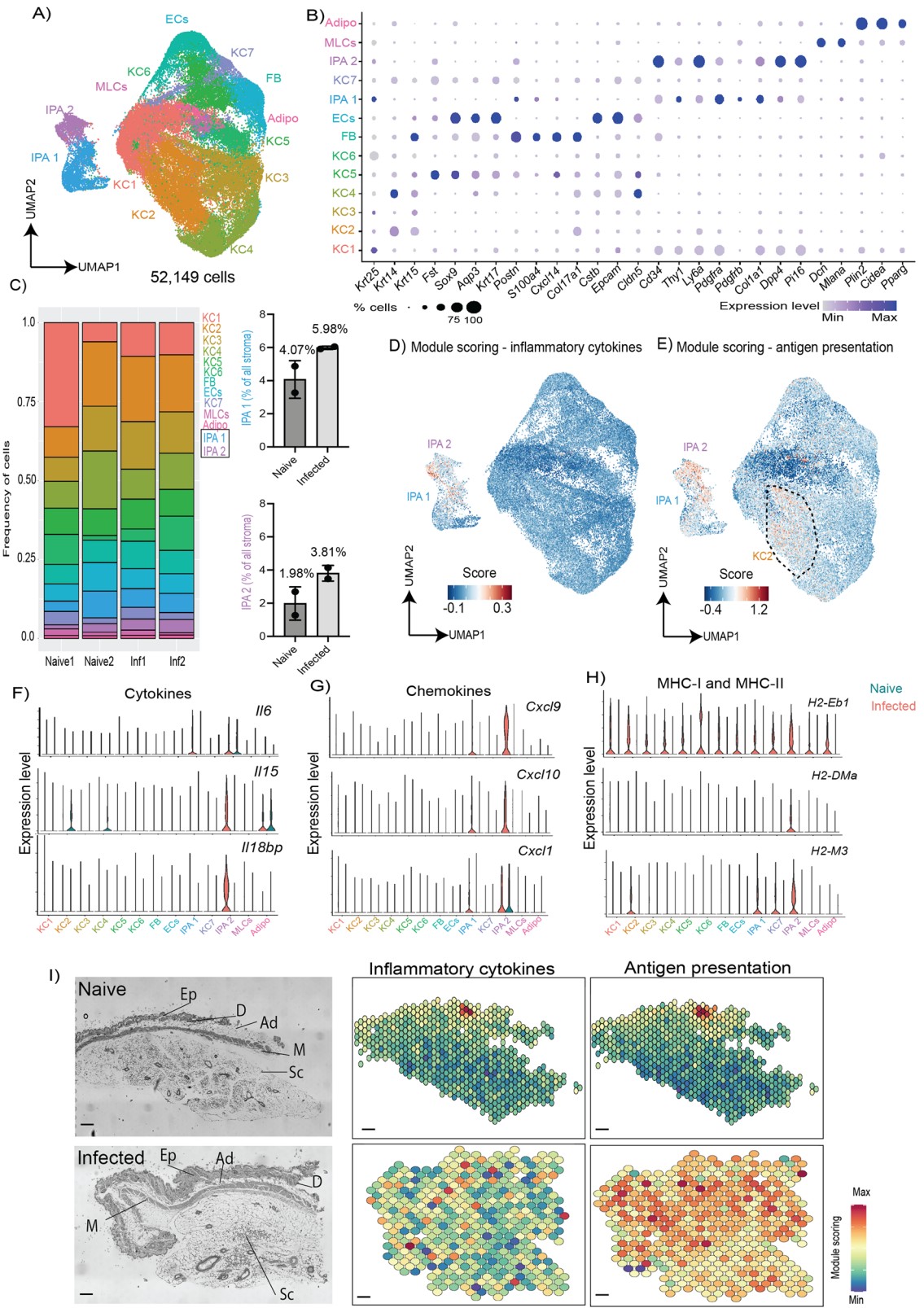

expansion of the γδT cells (Fig. 4D). Furthermore, ~50% of the cells within Vγ6+ cluster 1 express high levels of *Cd69*, whereas cells within Vγ6+ cluster 2 display low levels of *Cd69* expression, potentially suggesting that these two Vγ6+ clusters might represent γδT cells with different activation and/or trafficking phenotypes during infection. Spatial module scoring analysis predicted that Vγ6+ cells localise mainly to the dermis and epidermis of naïve mice (Fig. 4E), but their

localisation changes in response to infection, where they are found to be enriched in the subcutaneous adipose tissue (Fig. 4E). Indeed, we found Vγ6+ cells in the dermis and epidermis, as well as in the hypodermis of skin biopsies from naïve and infected animals from independent tissue sections using smFISH, validating our in silico predictions (Fig. 4I). Together, these results indicate that during infection there is an expansion of *Nrc1*+ NK cells and ILC2s, that are

**Fig. 2 | The murine skin stromal cells respond to infection by upregulating genes associated with antigen presentation and chemotaxis. A** Uniform manifold approximation and projection (UMAP) of 52,149 high-quality cells within the stromal subcluster, encompassing seven keratinocyte clusters (KC1-7), fibroblasts (FB), endothelial cells (ECs), melanocytes (MLCs), and interstitial preadipocytes (IPAs). **B** Dot plot representing the expression levels of top marker genes used to catalogue the diversity of skin stromal cells. The size of the dots represents the percentage of cells that express a given marker, and the colour intensity represents the level of expression. **C** Left panel: Frequency plot the different stromal cell types detected in the murine skin in naïve (*n* = 2 mice from the single cell experiment performed once) and infected (*n* = 2 mice from the single cell experiment performed once) samples. Module scoring for the overall expression of inflammatory cytokines. Right panel: Frequency of interstitial preadipocytes 1 (top) and 2 (bottom) in naïve and infected samples from the scRNAseq dataset. The data is presented as mean values +/− SD. Source data are provided as a Source Data file. Module scoring for Inflammatory cytokines (**D**) and genes associated with antigen presentation (**E**). Violin plot showing the expression level of several significant adipocyte-specific cytokines (**F**), chemokines (**G**), and major histocompatibility complex (MHC) I and II molecules (**H**). **I** Spatial module scoring for inflammation and antigen presentation in a naïve (top) and infected (bottom) skin section. Scale bar, 100 μm.

likely to contribute to the cytokine-mediated inflammation observed in the skin during infection. Similarly, our data demonstrates that there is a population of Vγ6⁺ cells within the subcutaneous adipose tissue compartment that exhibits features of local activation. During cold challenge, Vγ6⁺ cells are known to drive thermogenesis and lipid mobilisation *via* lipolysis through crosstalk with brown and beige adipocytes[31,32]. However, interactions between Vγ6⁺ cells and white adipocytes, in particular in the subcutaneous white adipose tissue during infection, have yet to be explored.

We next examined whether there is a cell-cell communication axis between adipocytes and all T cell clusters found in the skin during infection. NicheNet analysis indicates that adipocytes provide several critical cues for T cell recruitment and activation, including the activating factors *Cd40*, *Tnfsf18*, and *Icam1*, chemokines *Ccl12*, *Ccl8*, and cytokines *Tnf*, *Il10*, and *Il6* (Fig. 4G). These adipocyte-derived ligands are predicted to be sensed by skin-resident T cells *via* the expression of *Cd40lg*, *Itgax*, *Il10ra*, *Il6st*, *Tnfrsf1b*, and *Tnfrsf18*. In the spatial context, we noted that the expression of adipocyte derived *Tnfsf18* is restricted to the subcutaneous adipose tissue in the infected skin, coinciding with the expression of T cells expressing their cognate receptor, *Tnfrsf18* (Fig. 4H). We validated this pattern of expression using smFISH and found that *Tnfsf18* was upregulated in the parenchyma of the subcutaneous adipose tissue, coinciding with *Adipoq*⁺ adipocytes and *Tcrd*⁺ γδ T cells (Fig. 4I). Taken together, these results demonstrate that chronic skin infection with *T. brucei* leads to the local activation of a subpopulation of Vγ6⁺ cells in a process likely aided by subcutaneous adipocytes.

## Vγ6⁺ cells are essential for controlling skin inflammation, local CD8⁺ T cell activation, and subcutaneous adipose tissue wasting independently of skin-resident T_H1 T cells

So far, our data indicates that Vγ6⁺ cells expand significantly in the chronically infected skin, and we predicted that these populations receive activation signals from adipocytes within the subcutaneous adipose tissue. To better understand if the skin-resident Vγ6⁺ cells are involved in a communication axis with subcutaneous adipocytes, we re-clustered the T cells and adipocytes together to conduct ligand-receptor mediated cell-cell communication analyses between these cell types. We found that Vγ6⁺ cells express several key genes involved in driving mesenchymal (including preadipocyte) differentiation and changes in lipid metabolism (Fig. 5A). For instance, we detected an upregulation of the Cardiotrophin-like cytokine factor 1 (*Clcf1*) and Amphiregulin (*Areg*) in the Vγ6⁺ cells during infection, which is predicted to engage with their cognate receptors *Cntfr* and *Egfr*, respectively (Fig. 5A and B). Other Vγ6⁺ cell-derived factors promoting mesenchymal differentiation are *Cd24a*, the Placental growth factor (*Pgf*), Sialophorin (*Spn*), and Pleiotrophin (*Ptn*), which are predicted to engage with *Pparg*⁺ adipocytes *via* Selectin P (*Selp*), Neuropilin 1/2 (*Nrp1/Nrp2*), Syndecan 3 (*Sdc3*), and Sialic Acid Binding Ig Like Lectin 1 (*Siglec1*), respectively (Fig. 5A). Interestingly, some of these Vγ6⁺ cell-derived factors are able to drive lipid mobilisation[31–33], and together with our results this suggests that during infection Vγ6⁺ γδT cells may be involved in promoting lipolysis to mobilise energy storage from

adipocytes. Based on these observations, we hypothesised that, during infection, Vγ6⁺ cells expand in the skin and are important for driving subcutaneous adipose tissue wasting, limiting IFNγ-driven skin inflammation, and controlling parasite burden. To test our hypothesis, we infected mice with a double Vγ4/6 T cell knockout (on an FVB/N genetic background; Vγ4/6⁻/⁻) for a period of 21 days, and we monitored parasitaemia and clinical scores throughout infection. We observed that both Vγ4/6⁻/⁻ and FVB/NJ mice displayed the same levels of parasitaemia (Supplementary Fig. 4A) and parasite burden in the skin as measured by qRT-PCR against the trypanosome-specific *Prf2* gene and histological staining of BiP (Supplementary Fig. 4B and C). We also detected both slender and stumpy forms in all dermal layers in both infected Vγ4/6⁻/⁻ and FVB/NJ mice without significant differences between strains (Supplementary Fig. 5A), suggesting that Vγ4⁺ and Vγ6⁺ cells are dispensable for controlling parasite burden in the skin. Interestingly, there were no significant differences in the bodyweight of FVB/NJ and Vγ4/6⁻/⁻ mice over the course of infection (Fig. 5C), but we noted that the mass of the subcutaneous adipose tissue (normalised to bodyweight) was significantly reduced in the FVB/NJ mice but not in the Vγ4/6⁻/⁻ mice (Fig. 5D), indicating that Vγ4⁺ and Vγ6⁺ cells are involved in subcutaneous adipose tissue wasting. Furthermore, these effects seem to be restricted to the subcutaneous adipose tissue as we failed to detect significant differences in the spleen or gonadal white adipose tissue mass in response to infection between strains (Supplementary Fig. 5B).

To further understand the impact of Vγ4⁺ and Vγ6⁺ cells on adipose tissue responses to infection, we next examined histological sections of skin samples from infected Vγ4/6⁻/⁻ and FVB/NJ mice, as well as their counterpart naïve controls. Compared to infected FVB/NJ mice, the Vγ4/6⁻/⁻ mice displayed more severe signs of skin inflammation. Specifically, we observed higher follicular atrophy in the dermis and hypodermis, as well as diffuse lymphocyte aggregates containing a large number of plasma cells and oedema in the subcutaneous adipose tissue compared to infected FVB/NJ mice (Fig. 5E and Supplementary Data 3). Histological analysis indicated that, during infection, FVB/NJ mice experience reductions in subcutaneous adipocyte size compared with their naïve counterparts, consistent with previous reports in the trypanotolerant C57BL/6 background[24] (Fig. 5E and F). In contrast, morphometric analysis revealed that adipocytes on the Vγ4/6⁻/⁻ mice were significantly smaller in area in naïve animals compared to the FVB/NJ background (Fig. 5E−H), potentially highlighting a role for the Vγ4⁺ and Vγ6⁺ cells in maintaining adipocyte function under homeostasis. Moreover, our morphometric analyses revealed that adipocytes within the subcutaneous adipose tissue of infected FVB/NJ mice were significantly smaller than those in naïve mice, whereas infection did not significantly impact adipocyte size in the Vγ4/6⁻/⁻ mice (Fig. 5E−G).

Lastly, we hypothesised that loss of Vγ4⁺ and Vγ6⁺ cells may lead to an increased frequency of IFNγ⁺ T cells, which could explain the increased skin inflammation that we observed during infection. However, within the T cell compartment, we failed to detect significant differences in the frequency of skin-resident CD4⁺ and CD8⁺ T cells, or the frequency of IFNγ-producing CD4⁺ T cells (Supplementary Figs. 6

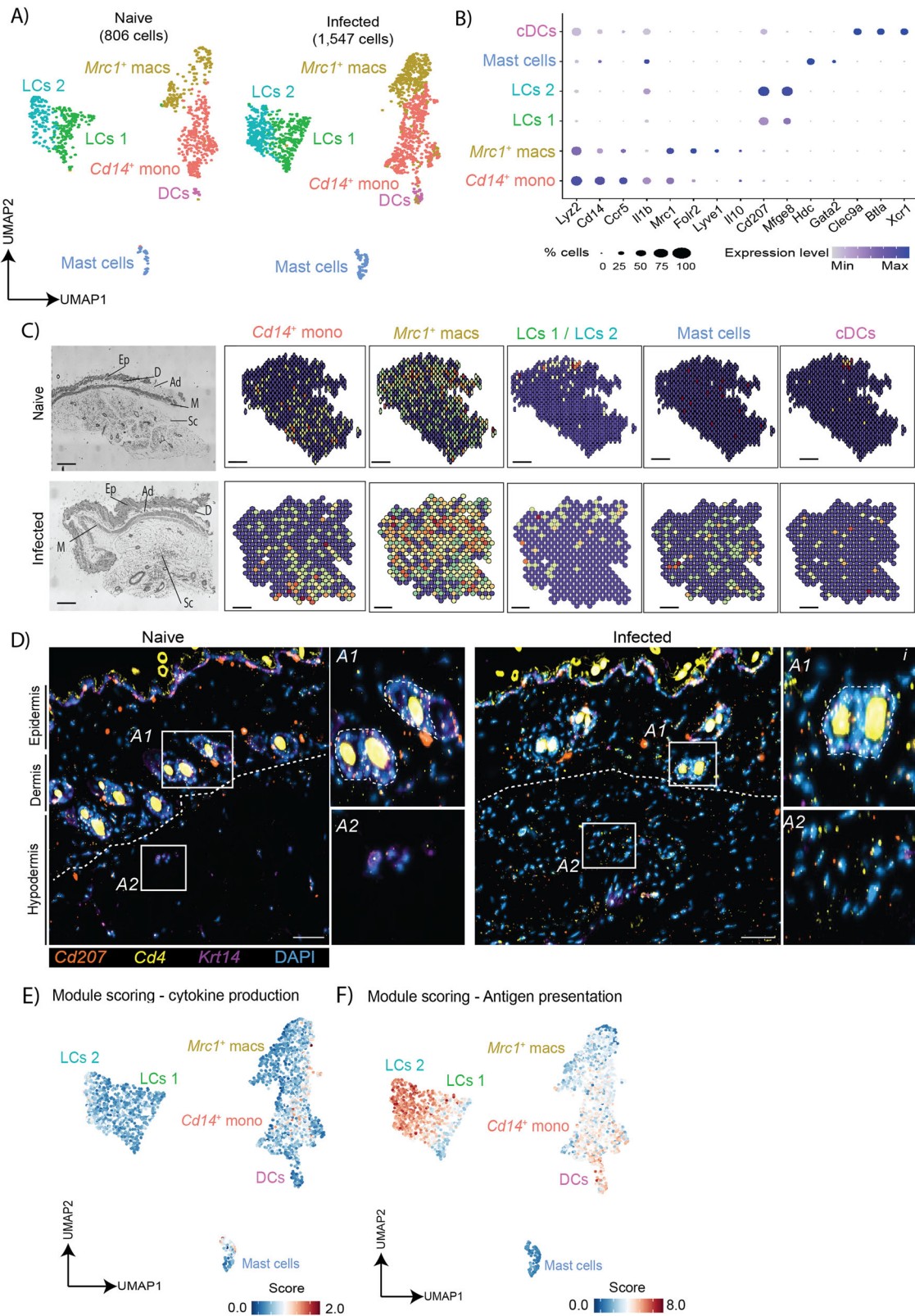

and 7), indicating a dispensable role for Vγ4⁺ and/or Vγ6⁺ cells in limiting T_H1-mediated T cell responses. As expected, within the CD8⁺ T cell compartment, we observed that *T. brucei* infection induced a significant increase in the frequency of activated dermal CD8⁺ T cells primed to produce IFNγ when challenged ex vivo compared to naïve controls (Supplementary Figs. 6 and 7). In the skin of the Vγ4/6⁻/⁻ mice, there was a significant increase in the frequency of activated IFNγ-

producing CD8⁺ T cells compared to FVB/N controls under homeostatic conditions, which were exacerbated during infection. This may suggest that Vγ4⁺ and/or Vγ6⁺ cells are important for restraining CD8⁺ T cells responses in the skin, either directly (e.g., controlling the activation threshold) or indirectly (e.g., through the recruitment/activation of other partners involved in CD8⁺ T cell activation) (Supplementary Figs. 6 and 7). Taken together, our results

**Fig. 3 | The murine skin is colonised by a myriad of myeloid cells during chronic _T. brucei_ infection. A** Uniform manifold approximation and projection (UMAP) of 2353 high-quality cells within the myeloid cluster were re-analysed to identify a total of four subclusters, including dendritic cells (DCs), mast cells, and two populations of macrophages (_Mrc1_⁺ macs and _Cd14_⁺ mono), and two populations of Langerhans cells (LCs 1 and LCs 2). **B** Dot plot representing the expression levels of top marker genes used to catalogue the diversity of myeloid cells. **C** Integration of single cell and spatial transcriptomics datasets for the myeloid cell identified during infection, including _Cd14_⁺ monocytes, _Mrc1_⁺ macrophages, _Cd207_⁺ Langerhans cells, mast cells, and conventional dendritic cells (cDCs). Scale bar, 100 μm. The corresponding histological section is included on the left, including an annotation of epidermis

(Ep), dermis (D), adipose tissue (Ad), muscle (M), and subcutaneous adipose tissue (Sc). **D** Representative smFISH targeting _Cd207_ (orange), _Cd4_ (yellow), and _Krt14_ (purple) around the perifollicular space (denoted as A1; The hair follicles are highlighted with dotted lines) and in the adipose tissue in the hypodermis (denoted as A2) in skin biopsies from independent naïve and infected animals. Scale bar, 50 μm. The results presented here are representative from two independent experiments. The images have been adjusted for brightness and contrast to increase resolution. Please note that the sections contain hair with high auto-fluorescence. Module scoring for the overall expression of inflammatory cytokines (**E**) and genes associated with antigen presentation (**F**).

demonstrate that Vγ4⁺ and/or Vγ6⁺ cells are critical for limiting skin pathology, as well as driving subcutaneous adipose tissue wasting, in a process likely involving IL-17 signalling.

## Discussion

To address key questions about the immune response of the skin to chronic infection with _T. brucei_, this study aimed to characterise changes in skin cell populations using single cell transcriptomics, as well as to determine how the cell populations detected by single cell transcriptomics are distributed throughout the skin during infection using spatial transcriptomics. With this information, we then modelled cell-cell interactions in the skin during _T. brucei_ infection, to understand immune-stromal crosstalk and how this influences the immune response to infection. Here, using a combination of cutting-edge technologies and genetic murine models, we demonstrated that IL-17-producing Vγ6⁺ cells play a critical role in controlling skin inflammation (Fig. 6). Furthermore, our data highlight a previously unappreciated interaction between subcutaneous interstitial preadipocytes and mature adipocytes and skin-dwelling T cells (including γδ T cells), which mediate T cell responses and subcutaneous adipose tissue wasting (Fig. 6).

We first generated a spatially resolved single cell atlas of the murine skin during chronic _T. brucei_ infection. From these analyses, several observations are worth discussing in detail. First, we observed significant changes in the skin stromal and immune compartment without the formation of granulomatous lesions, indicative of sub-clinical inflammatory processes when compared to naïve controls. Using module scoring of inflammatory cytokines and chemokines, as well as genes associated with antigen presentation, we detected inflammatory signatures predominantly in populations of _Cd14_⁺ monocytes, Langerhans cells, and interstitial preadipocytes located in the subcutaneous adipose tissue.

Following a reclustering of stromal cells, we identified two populations of _Dpp4_⁺ interstitial preadipocytes (IPA1 and IPA2) that upregulate inflammatory cytokines, chemokines, and molecules associated with antigen presentation. The chemokines _Cxcl1_, _Cxcl9_ and _Cxcl10_ were upregulated in both populations of interstitial preadipocytes but were higher in IPA2. These chemokines are secreted to recruit neutrophils[34], CD4⁺ and γδ T cells[35,36], and natural killer cells[37], and their upregulation exclusively by preadipocytes suggests that these cells are critical drivers of immune recruitment to the skin during _T. brucei_ infection. Supporting this, in the skin of infected mice we found expansion of CD4⁺ T cells, γδ T cells, and NK cells, and interestingly this may represent a feedback loop whereby these immune cells suppress differentiation of preadipocytes to mature adipocytes, as observed using in vitro models[38]. In addition to preadipocytes, we also found that mature adipocytes upregulate chemokines during infection, including _Ccl8_ and _Ccl12_, which are drivers of monocyte recruitment[39]. Moreover, these cells upregulated _Il6_ and _Il10_, which were predicted to communicate with T cells through _Il6st_ and _Il10ra_, respectively. To our knowledge, although adipose tissue immune populations are known to express IL-10[40], and adipocytes express the IL-10 receptor[41], this is the first time that adipocyte _Il10_ expression has

been reported. In the subcutaneous adipose tissue, IL-10 signalling limits energy expenditure and lipolysis in mouse models of cold exposure and obesity[41], but its effects on adipocytes during infection remain unknown. Our observations may suggest that during _T. brucei_ infection, IL-10 acts in both an autocrine and paracrine fashion, whereby adipocytes secrete the cytokine and it suppresses adipose tissue lipolysis, whilst simultaneously suppressing CD4⁺ T cell activity[42]. Conversely, IL-6 is a driver of lipolysis and fatty acid oxidation[43] and is associated with weight loss and fat wasting in diseases such as HIV[44] and cancer[45]. It is, therefore, unclear how these two cytokines impact adipocyte activity when both are present.

Importantly, we also identified a population of skin-resident Vγ6⁺ T cells that expand in response to skin infection. These Vγ6⁺ cells are primed to produce IL-17A and IL-17F during development in the thymus[46,47], and upon maturation they migrate to multiple tissues throughout the body, including the skin, where they become resident immune cells, offering a first line of response to infection[48]. The γδ T cells that we identified in our dataset express markers putatively associated with activation, including _Cd44, Cd69_, and _Nr4a1_, potentially suggesting the existence of local drivers of γδ T cell activation in the infected skin. Both in silico predictions and in vivo analyses indicate that these γδ T cells are likely to be IL-17A-producing Vγ6⁺ cells based on the expression levels of _Tcrg-C1, Cd163l1_, and _Cd27_, as previously reported[30]. We also observed two populations of Vγ6⁺ cells, which we hypothesise represent "resting" and "activated" populations, based on the expression of _Cd44, Cd69_, and _Nr4a1_. However, unlike other γδ T cell populations in the skin, such as dendritic epidermal T cells that reside solely in the dermis, Vγ6⁺ cells may be able to recirculate between tissues[49,50]. In this scenario, our data may provide evidence of one Vγ6⁺ population exiting the skin and a separate population entering the skin during infection. However, future studies are required to dissect the migratory dynamics of skin Vγ6⁺ cells in the context of skin infection.

Our findings further highlight the importance of IL-17A signalling in the skin of mice infected with _T. brucei_, consistent with our previous work proposing IL-17A as a critical driver of subcutaneous and inguinal adipose tissue wasting[24]. Interestingly, in the spatial context, these IL-17A-producing Vγ6⁺ cells are located in the subcutaneous adipose tissue layer of the skin and are predicted to establish crosstalk with adipocytes _via_ several molecules, including T cell co-stimulatory signals such as _Tnfsf18_, which engages with GITR (_Tnfrsf18_) to lower the T cell activation threshold[51]. These cells also express _Cd40_, indicating previously unappreciated crosstalk between stromal adipocytes and Vγ6⁺ cells during _T. brucei_ infection in the skin. Consistent with this, mice lacking Vγ4/6⁺ T cells display a higher number of plasma cells and more severe skin inflammation compared to wild type controls. Mice deficient in Vγ4⁺ and Vγ6⁺ cells are known to develop increased numbers of plasma cells and spontaneous germinal centre formation[52], which may dysregulate the immune response to infection. We found that the increased inflammation in the skin of infected Vγ4/6⁻/⁻ mice may be mediated, at least in part, by controlling the activation threshold of skin-resident CD8⁺ T cells. The increased capacity of CD8⁺ T cells to produce IFNγ in the skin of naïve Vγ4/6⁻/⁻ mice suggests that these cells

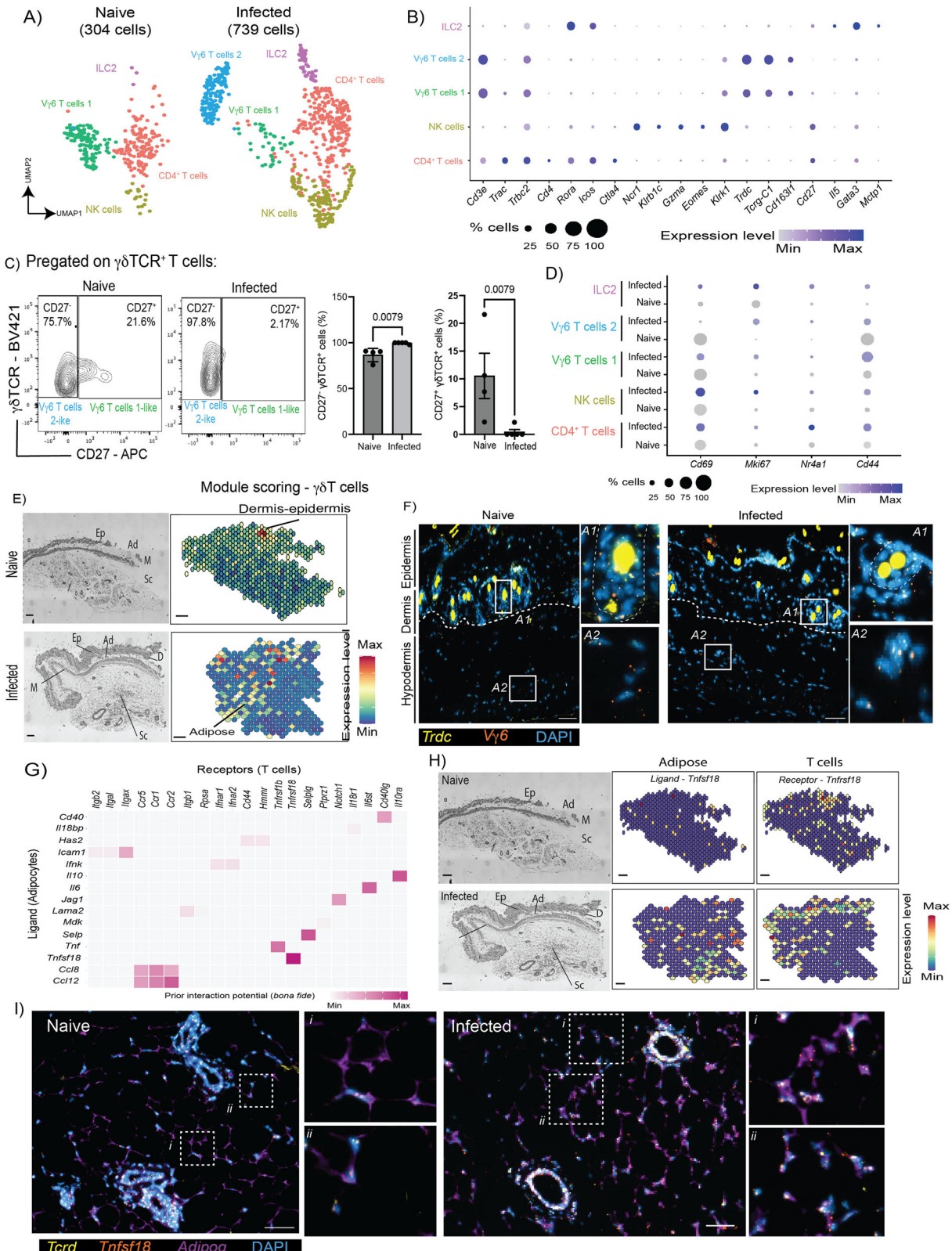

play a role in constraining CD8$^+$ T cell activity under homeostasis. Indeed, the potential capacity of dermal γδ T cells (including Vγ6$^+$ cells) to limit pathogenic CD8$^+$ T responses might be associated with the expression of immunomodulatory mediators such as PD-1 (*Pdcd1*)[53] and Galectin-1 (*Lgals1*)[54], the latter being highly expressed in the cells within Vγ6$^+$ cells cluster 1 in our dataset. An alternative possibility is that the increased severity of skin inflammation in infected Vγ4/6$^{-/-}$

mice is due to exacerbated recruitment of neutrophils, as reported in metastatic breast cancer[55]. These hypotheses remain to be explored in more detail in future studies.

Strikingly, we found that Vγ4/6$^{-/-}$ mice do not lose subcutaneous adipose tissue to the same extent as wild type controls during infection, mirroring our previous studies where we proposed IL-17A as a driver of infection-associated adipose tissue wasting[24]. This was limited

**Fig. 4 | Chronic *T. brucei* infection triggers the activation of dermal Vγ6⁺ cells.**
**A** Uniform manifold approximation and projection (UMAP) of 1043 T cells from naïve and infected skin samples. We detected a total of five subclusters including type 2 innate lymphoid cells (ILC2s), *Cd4*⁺ T cells, natural killer (NK) cells, and γδT cells. **B** Dot Plot depicting the expression level of top marker genes for the skin T cell subcluters. **C** Left: Flow cytometry of CD27⁻ and CD27⁺ γδT cells in murine skin infected (*n* = 5 mice examined from two independent experiments) with *T. brucei* and naïve controls (*n* = 4 mice examined over two independent experiments). Right: Quantification of flow cytometry data. Statistical analysis was conducted using a parametric two-sided *T* test. A *p* value < 0.05 was considered significant. The data is presented as mean values +/− SD. Source data are provided as a Source Data file. **D** Dot Plot depicting the expression level of top genes associated with T cell activation and TCR engagement for the skin T cell subcluters. **E** Spatial module scoring for γδ T cell in naïve and infected skin biopsies used for spatial transcriptomics. Scale bar, 100 μm. **F** Representative smFISH

(two independent experiments) targeting *Trdc* (yellow) and *Vγ6* (orange*)* around the perifollicular space (denoted as A1) and in the adipose tissue in the hypodermis (denoted as A2) in skin biopsies from naïve and infected animals. The area corresponding to the hair follicle is denoted with dotted lines in A1. Please note that the sections contain hair follicles with high autofluorescence. Scale bar, 50 μm. **G** In silico cell-cell interaction analysis between adipocytes ("senders") and T cells ("receivers"). The heatmap is colour coded to represent the strength of the interaction. **H** Spatial feature plot depicting the expression of adipose-derived ligand *Tnfsf18* and T-cell specific receptor *Tnfrsf18*. Scale bar, 100 μm. **I** Representative smFISH (from two independent experiments) targeting *Trdc* (yellow), *Tnfsf18* (orange), and *Adipoq* (purple) in the adipose tissue in the hypodermis in skin biopsies from naïve and infected animals. Two areas showing co-localisation of all the probes are highlighted for reference. The images have been adjusted for brightness and contrast to increase resolution. Scale bar, 50 μm.

to the subcutaneous white adipose tissue, as the gonadal white adipose tissue was equally wasted in both wildtype and Vγ4/6⁻/⁻ mice, potentially arguing in favour of functional differences between white adipose tissue depots. In both wild type and Vγ4/6⁻/⁻ mice we observed comparable parasite tissue burden, with both slender and stumpy developmental forms of the parasite readily detected in all the dermal layers of these mice. Furthermore, we failed to detect significant differences in the frequency of skin-resident T$_H$1 T cells, which strongly suggests that both parasites and T$_H$1 T cells are dispensable for driving subcutaneous adipose tissue wasting, which is typically observed in this experimental infection setting[24,56]. Thus, it is tempting to speculate that IL-17A-producing Vγ6⁺ cells (and potentially other sources of IL-17A such as T$_H$17 T cells) promote subcutaneous adipose tissue lipolysis to fuel an efficient immune response against the parasites, although the molecular mechanisms underlying this process need to be investigated in more detail. However, in the study presented here, we were unable to dissect the relative contribution of Vγ4⁺ or Vγ6⁺ cells to skin inflammation or the interactions between different γδ T cell subsets that reside in the skin under both homeostatic conditions and inflammation.

Our data strongly indicate that the subcutaneous adipose tissue is an active site for immune priming and activation, placing the adipocytes at the core of this process. Thus, in the context of chronic skin infection with *T. brucei*, we propose a model whereby subcutaneous adipocytes (in addition to Langerhans cells and keratinocytes) have a critical role as coordinators of local innate and adaptive immune responses. In the context of trypanosome infection, subcutaneous adipocytes may detect the presence of parasites (e.g., *via* Toll-like receptor signalling) to trigger the recruitment and activation of innate immune cells such as γδ T cells to mobilise energy stores to meet the energetic requirements needed to control infection, as recently proposed[57]. In this manuscript, we focus on characterising the immunological events that the parasites encounter in the skin prior to forward transmission (host to vector). However, given the immunomodulatory nature of the salivary components delivered at the site of inoculation by tsetse flies[58–61], it would also be interesting to explore the dynamics of γδ T cell activation and adipocyte responses in the skin during the onset of infection.

Together, our spatially resolved atlas of the murine skin during *T. brucei* infection offers a resource to the community interested in understanding how chronic infections affect skin homeostasis and immunity. It is important to acknowledge that although robust, the current strategies to deconvolve cell type distribution at the spot level for spatial transcriptomics (for example, *via* anchor-based integration approaches such as Seurat, or other models such as those provided by Giotto[62]) rely heavily on the diversity of the single cell atlas used as an input. In our study, we offset this limitation by performing a series of additional validations, including in situ hybridisation and flow cytometry, of cell types and tissue regions of interest in vivo. We anticipate

that as the field of spatial biology moves forward, additional packages and spot deconvolution strategies will be refined to capture additional nuances of the dataset. Future work is required to examine the consequences of infection on adipocyte differentiation and function, and whether these processes are directly controlled by γδ T cells, as shown during cold exposure[31,32]. Furthermore, our dataset provides strong evidence for an engagement of several cell types within the stromal compartment in the skin in response to infection and may suggest that the efficacy, robustness, and timing of the local immune response may be determined by cell types traditionally associated with non-immunological functions such as mesenchymal cells (including interstitial preadipocytes). We envision that future work dissecting the role of these various cell types and communication axis will address some of the fundamental questions arising from this study.

## Methods

### Murine infections with *Trypanosoma brucei*

6–8-week-old wild-type female BALB/c (stock 000651) and FVB/NJ mice (stock 001800) were purchased from Jackson Laboratories. Vγ4/6⁻/⁻ mice (a gift from Rebecca O'Brien, National Jewish Health) were backcrossed to FVB/NJ. Female mice aged (6–8 weeks old) were used for infection. Six-to-eight-week-old wild-type female BALB/c (stock 000651) and FVB/NJ mice (stock 001800) were purchased from Jackson Laboratories. For infections, mice were inoculated by intraperitoneal injection with ~2 × 10³ parasites of strain *T. brucei brucei* GVR35. Parasitaemia was monitored by regular sampling from tail venepuncture and blood was examined using phase microscopy and the rapid "matching" method[16]. Uninfected mice of the same strain, sex and age served as uninfected controls. Mice were fed *ad libitum* and kept on a 12 h light–dark cycle, room temperature between 20 °C–24 °C and humidity between 50–70%. All the experiments were conducted in the morning, between 8 h and noon.

### Skin tissue processing and preparation of single cell suspension for single-cell RNA sequencing

Infected animals and naïve controls were anesthetized with isoflurane at 21 days post-infection and perfused transcardially with 25–30 ml of ice-cold 1X PBS containing 0.025% (wt/vol) EDTA. Skin biopsies from the abdominal flank area were harvested. Some sections were also placed in 4% PFA for 16 h prior to embedding in paraffin and histological analysis. Single-cell dissociations for scRNAseq experiments were performed using the 2-step protocol as previously published[63]. Briefly, excised 4 cm² skin sections were minced into small pieces with a scalpel blade and enzymatically digested with Collagenase type I (500 U/ml; Gibco) and DNAse I (1 mg/ml; Sigma) in HBSS containing 0.04% BSA (Invitrogen) for ~1 h at 37 °C with shaking at 300 rpm. Liberated cells from the partially digested tissue were pushed through a 70 μm nylon mesh filter with an equal volume of HBSS 0.04% BSA, and then kept on ice. The tissue remaining in the 70 μm filter was incubated

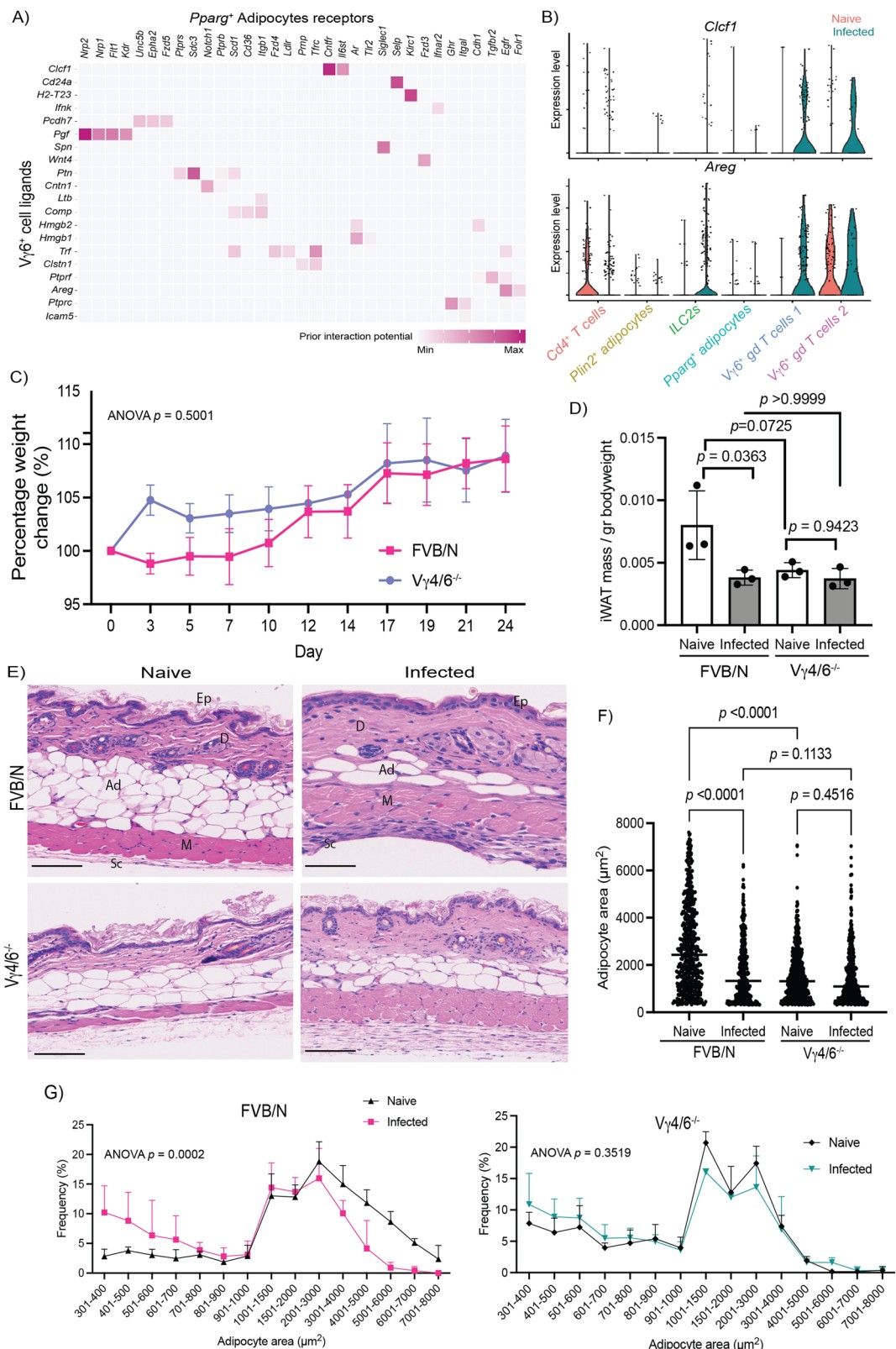

in 0.05% trypsin EDTA for 15 min at 37 °C, and then liberated cells were pushed through the filter with an equal of HBSS 0.04% BSA. Flowthrough from the two digestion steps was combined, passed through a 40 μM filter to remove any cell aggregate and spun at 350 g for 10 min at 4 °C. Finally, cells were passed through a MACS dead cell removal kit (Miltenyi Biotec) and diluted to ~1000 cells/μl in 200 μl HBSS 0.04% BSA and kept on ice until single-cell capture using the 10X Chromium

platform. The single cell suspensions were loaded onto independent single channels of a Chromium Controller (10X Genomics) single-cell platform. Briefly, ~25,000 single cells were loaded for capture using 10X Chromium NextGEM Single cell 3 Reagent kit v3.1 (10X Genomics). Following capture and lysis, complementary DNA was synthesized and amplified (12 cycles) as per the manufacturer's protocol (10X Genomics). The final library preparation was carried out as recommended

**Fig. 5 | Vγ6⁺ cells are essential for controlling skin inflammation and subcutaneous adipose tissue wasting independently of skin-resident T_H1 T cells.**
**A** In silico cell-cell interaction analysis between Vγ6⁺ cells ("senders") and *Pparg*⁺ adipocytes ("receivers") based on the upregulation of ligand-receptor pairs. The heatmap is colour coded to represent the strength of the interaction. **B** Expression level of *Clcf1* and *Areg*, two of the most significant upregulated Vγ6⁺ cells-derived ligands predicted to interact with subcutaneous adipocytes. **C** Bodyweight of FVB/NJ and Vγ4/6⁻/⁻ mice over the course of infection (*n* = 3 mice/group). A non-parametric, one-way ANOVA was used to determine the level of significance. A *p* value < 0.05 is considered significant. Source data are provided as a source data file. **D** Subcutaneous adipose tissue mass from naïve and infected FVB/NJ and Vγ4/6⁻/⁻ mice normalised to whole bodyweight. A non-parametric, one-way ANOVA was used to determine the level of significance. A *p* value < 0.05 is considered significant. Source data are provided as a Source data file. **E** Representative H&E staining from skin biopsies obtained from FVB/NJ and Vγ4/6⁻/⁻ naïve and infected

mice, from two independent experiments. Scale bar: 100 μm. epidermis (Ep), dermis (D), adipose tissue (Ad), muscle (M), subcutis (Sc). **F** Analysis of mean adipocyte area (μm²) in naïve and infected FVB/NJ and Vγ4/6⁻/⁻ mice. *n* = 5 biological replicates per group over two independent experiments, from two independent experiments. Lipid droplets were measured from 3 distinct areas in each image and then combined for each biological replicate. A non-parametric, one-way ANOVA was used to determine the level of significance. A *p* value < 0.05 is considered significant. Source data are provided as a Source data file. **G** Frequency plot of the adipocyte area represented in (**D**) for naïve and infected FVB/NJ (left panel) and Vγ4/6⁻/⁻ mice (right panel). *n* = 5 biological replicates per group, from two independent experiments. Lipid droplets were measured from 3 distinct areas in each image and then combined for each biological replicate. The data is presented as mean +/− SD. A non-parametric, one-way ANOVA was used to determine the level of significance. A *p* value < 0.05 is considered significant. Source data are provided as a Source data file.

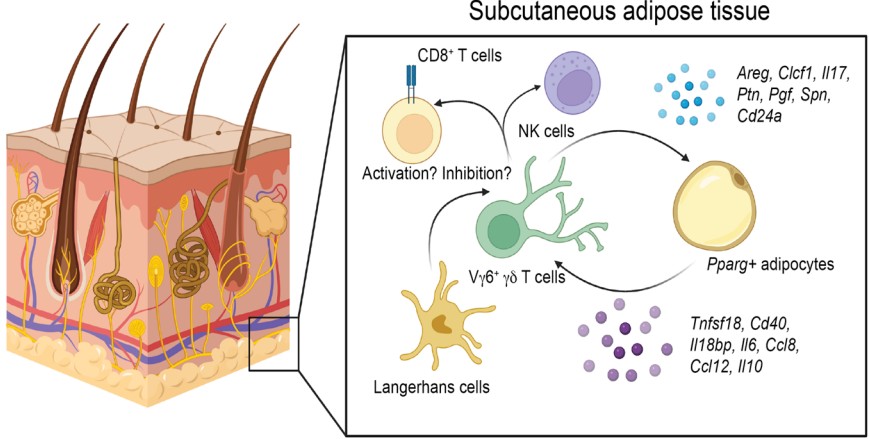

**Fig. 6 | Proposed model of stromal-immune interactions in the skin during *T. brucei* infection.** Based on our spatially-resolved single cell atlas, we propose a model whereby Vγ6⁺ cells act concertedly with *Pparg*⁺ adipocytes (and potentially preadipocytes) to coordinate local immune responses, possibly *via* the recruitment of immune cells (e.g., CD8⁺ T cells and natural killer (NK) cells). The *Pparg*⁺

adipocytes in this context provide important cues for T cell activation that we hypothesise might be involved in triggering Vγ6⁺ cells -mediated responses. Other stromal cells such as Langerhans cells and keratinocytes are also likely to be involved in this process *via* antigenic presentation. Created with Biorender (Agreement number WT25PETNWV).

by the manufacturer with a total of 14 cycles of amplification. The amplified cDNA was used as input to construct an Illumina sequencing library and sequenced on a Novaseq 6000 sequencer by Glasgow Polyomics.

### Read mapping, data processing, and integration

For FASTQ generation and alignments, Illumina basecall files (*.bcl) were converted to FASTQs using bcl2fastq. Gene counts were generated using Cellranger v.6.0.0 pipeline against a combined *Mus musculus* (mm10) and *Trypanosoma brucei* (TREU927) transcriptome reference. After alignment, reads were grouped based on barcode sequences and demultiplexed using the Unique Molecular Identifiers (UMIs). The mouse-specific digital expression matrices (DEMs) from all six samples were processed using the R (v4.2.1) package Seurat v4.1.0[17]. Additional packages used for scRNAseq analysis included dplyr v1.0.7[18], RColorBrewer v1.1.2 (http://colorbrewer.org), ggplot v3.3.5[19], and sctransform v0.3.3[20]. We initially captured 65,734 cells mapping specifically against the *M. musculus* genome across all conditions and biological replicates, with an average of 29,651 reads/cell and a median of 1762 genes/cell (Supplementary Data 1A and Supplementary Fig. 3B). The number of UMIs was then counted for each gene in each cell to generate the digital expression matrix (DEM). High quality cells were identified according to the following criteria: (i) nFeature >100 or <5000, (ii) nCounts >100 or <20,000, (iii) >30% reads mapping to mitochondrial genes, and (iv) >40% reads mapping to ribosomal genes, (v) genes detected <3 cells. After applying this cut-off, we

obtained a total of 56,876 high quality mouse-specific cells with an average of 29,651 reads/cells and a median of 1683 genes/cell (Supplementary Data 1A). The DoubletFinder packaged[64] (pN = 0.0675, pK = 0.01, using the *paramSweep* function to identify optimal values) identified <4% of predicted doublets (Supplementary Fig. 1C and Supplementary Data 1A). High-quality cells were then normalised using the *SCTransform* function, regressing out for total UMI and gene counts, cell cycle genes, and highly variable genes identified by both Seurat and Scater packages, followed by data integration using *IntegrateData* and *FindIntegrationAnchors*. For this, the number of principal components was chosen using the elbow point in a plot ranking principal components and the percentage of variance explained (15 dimensions) using a total of 5000 genes, and SCT as the normalisation method.

### Cluster analysis, marker gene identification, and subclustering

The integrated dataset was then analysed using *RunUMAP* (10 dimensions), followed by *FindNeighbors* (10 dimensions, reduction = "pca") and *FindClusters* (resolution = 0.3). The resolution used for these analyses was selected using the Clustree package[65] (Supplementary Fig. 1). With this approach, we identified a total of 16 cell clusters. The cluster markers were then found using the *FindAllMarkers* function (logfc.threshold = 0.25, assay = "RNA"). To identify cell identity confidently, we employed a supervised approach. This required the manual inspection of the marker gene list followed by an assignment of cell identity based on the expression of putative marker genes expressed

in the unidentified clusters. A cluster name denoted by a single marker gene indicates that the chosen candidate gene is selectively and robustly expressed by a single cell cluster and is sufficient to define that cluster (e.g., *Col1a1*, *Cd3g*, *Pparg*, *Krt14*, among others). When manually inspecting the gene markers for the final cell types identified in our dataset, we noted the co-occurrence of genes that could discriminate two or more cell types (e.g., T cells, fibroblasts). To increase the resolution of our clusters to help resolve potential mixed cell populations embedded within a single cluster, we subset fibroblasts, myeloid cells, and T cells and analysed them separately using the same functions described above. In all cases, upon subsetting, the resulting objects were reprocessed using the functions *FindVariableFeatures, ScaleData, RunUMAP, FindNeighbors*, and *FindClusters* with default parameters. The number of dimensions used in each case varied depending on the cell type being analysed but ranged between 5 and 10 dimensions. Cell type-level differential expression analysis between experimental conditions was conducted using the *FindMarkers* function (*min.pct* = 0.25, *test.use* = Wilcox) and (*DefaultAssay* = "SCT"). Cell-cell interaction analysis mediated by ligand-receptor expression level was conducted using NicheNet[63] with default parameters using "mouse" as a reference organism, comparing differentially expressed genes between experimental conditions (*condition_oi* = "Infected", *condition_reference* = "Uninfected"). Module scoring was calculated using the *AddModuleScore* function to assign scores to groups of genes of interest (*Ctrl* = 100, *seed* = NULL, *pool* = NULL), and the scores were then represented in feature plots. This tool measures the average expression levels of a set of genes, subtracted by the average expression of randomly selected control genes.

**10X Visium spatial sequencing library preparation and analysis**
**Tissue processing and library preparation.** Mice were shaved and dissected skin was stored in 4% paraformaldehyde for 24 h, before paraffin embedding. RNA was purified from FFPE sections to measure integrity, using the Absolutely Total RNA FFPE Purification Kit (Agilent Technologies) as per the manufacturer's instructions. RNA integrity was then measured by BioAnalyzer (Agilent Technologies) using the RNA 6000 Pico Kit (Agilent Technologies). Samples selected for sequencing all had a DV200 > 50%. We then placed 10 μm sections within the capture areas of Visium Spatial Slides (10X Genomics) and proceeded to perform Haematoxylin and Eosin (H&E) staining and image capture, as per the manufacturer's instructions. To deparaffinise, slides were incubated at 60 °C for 2 h before incubating twice in xylene at room temperature, for 10 min each, three times in 100% ethanol for 3 min each, twice in 96% ethanol for 3 min each, 85% ethanol for 3 min, and then submerged in Milli-Q water until ready to stain. For H&E staining, slides were incubated with Mayer's haematoxylin Solution (Sigma-Aldrich), Bluing Buffer (Dako) and Alcoholic Eosin solution (Sigma-Aldrich), with thorough washing in ultrapure water between each step. Stained slides were scanned under a microscope (EVOS M5000, Thermo). De-crosslinking was performed by incubating sections twice with 0.1 N HCl for 1 min at room temperature. Sections were then incubated twice with Tris-EDTA (TE) buffer pH 9.0 before incubating at 70 °C for 1 h. The Visium Spatial Gene Expression Slide Kit (10X Genomics) was used for reverse transcription and second strand synthesis, followed by denaturation, to allow the transfer of the cDNA from the slide to a collection tube. These cDNA fragments were then used to construct spatially barcoded Illumina-compatible libraries using the Dual Index Kit TS Set A (10x Genomics) was used to add unique i7 and i5 sample indexes, enabling the spatial and UMI barcoding. The final Illumina-compatible sequencing library underwent paired-end sequencing (150 bp) on a NovaSeq 6000 (Illumina) at GenomeScan.

After sequencing, FASTQ files were aligned to a merged reference transcriptome combining the *Mus musculus* genome (mm10). After alignment, reads were grouped based on spatial barcode sequences and demultiplexed using the UMIs, using the SpaceRanger pipeline version 1.2.2 (10X Genomics). Downstream analyses of the expression matrices were conducted using the Seurat pipeline for spatial RNA integration (Hao et al., 2021b; Stuart et al., 2019) (Supplementary Fig. 3A). Specifically, the data was scaled using the *SCTransform* function with default parameters. We then proceeded with dimensionality reduction and clustering analysis using *RunPCA* (assay = "SCT"), *FindNeighbours* and *FindClusters* functions with default settings and a total of 30 dimensions. We then applied the *FindSpatiallyVariables* function to identify spatially variable genes, using the top 1000 most variable genes and "markvariogram" as the selection method. To integrate our skin scRNAseq with the 10X Visium dataset, we used the *FindTransferAnchors* function with default parameters, using SCT as normalization method. Then, the *TransferData* function (weight.reduction = "pca", 30 dimensions) was used to annotate skin regions based on transferred anchors from the scRNAseq reference datasets. The resulting object was then scaled using the functions *ScaleData, FindVariableFeatures* (selection.method = "disp"), RunUMAP with default settings and a total of 15 dimensions. Spatially resolved expression of ligand-receptor pairs were then identified using the *FindAllMarkers* function (min.pct = 0.25, test.use = "roc"). For visualisation, we used the *SpatialFeaturePlot* function with default parameters and min.cutoff = "q1".

For single cell and spatial transcriptomics data integration, we used the anchor-based integration workflow in the Seurat package vignette and the Giotto package[62], with default parameters (30 dimensions, resolution = 0.4, k = 15, n_iterations = 1000). Briefly, after quality control and filtering, spatial transcriptomic samples were normalised independently, employing variance stabilisation through the *SCTransform* function with default parameters. We then identified anchors to be transferred between datasets using the function *FindTransferAnchors* and *TransferData* with default parameters and a total of 30 dimensions.

**Single molecule fluorescence in situ hybridisation (smFISH) using RNAscope.** smFISH experiments were conducted as follows. Briefly, to prepare tissue sections for smFISH, infected animals and naïve controls were anesthetized with isoflurane, and skin sections from the flank were dissected and placed in 4% PFA overnight prior to embedding in paraffin. Paraffin-embedded skin sections (3 μm) were prepared and placed on SuperFrost Plus microscope slides. Sections were then dehydrated in 50, 70 and 100% ethanol. RNAscope 2.5 Assay (Advanced Cell Diagnostics) was used for all smFISH experiments according to the manufacturer's protocols. We used the following probes: *Tbr-Gapdh* (Cat # 1103198-C1), *Tbr-Pyk1* (Cat # 1103208-C2), *Tbr-Pad2* (Cat #1103218-C3), *Tbr-Ep1* (Cat # 1103221-C4), *Cd207* (Cat # 452521-C1), *Trdc* (Cat # 449351-C4), *Cd4* (Cat # 406841-C4), *Krt14* (Cat # 422521-C2), *Tcrg-V6* (Cat # 495751-C1), *Tnfsf18* (Cat # 408491-C1), *Adipoq* (Cat # 440051-C3). All RNAscope smFISH probes were designed and validated by Advanced Cell Diagnostics. For image acquisition, 16-bit laser scanning confocal images were acquired with a 63x/1.4 plan-apochromat objective using an LSM 710 confocal microscope fitted with a 32-channel spectral detector (Carl Zeiss). Lasers of 405 nm, 488 nm and 633 nm excited all fluorophores simultaneously with corresponding beam splitters of 405 nm and 488/561/633 nm in the light path. 9.7 nm binned images with a pixel size of 0.07 μm × 0.07 μm were captured using the 32-channel spectral array in Lambda mode. Single fluorophore reference images were acquired for each fluorophore and the reference spectra were employed to unmix the multiplex images using the Zeiss online fingerprinting mode. smFISH images were acquired with minor contrast adjustments as needed, and converted to grayscale, to maintain image consistency.

**Histological analysis of adipocyte size.** The skin was placed into 4% paraformaldehyde (PFA) and fixed overnight at room temperature.

PFA-fixed skin biopsies were then embedded in paraffin, sectioned, and stained by the Veterinary Diagnostic Services facility (University of Glasgow, UK). Sections were H&E stained for adipocyte size analysis. Slide imaging was performed by the Veterinary Diagnostic Services facility (University of Glasgow, UK) using an EasyScan Infinity slide scanner (Motic, Hong Kong) at 20X magnification. To determine subcutaneous adipocyte size in skin sections, images were first opened in QuPath (v. 0.3.2)[66], before selecting regions and exporting to Fiji[67]. In Fiji, images were converted to 16-bit format, and we used the Adiposoft plugin[68] to quantify adipocyte size within different sections.

**Semi-quantitative evaluation of the parasite burden and inflammation in skin sections.** Paraffin-embedded skin samples were cut into 3 μm sections and stained for *T. brucei* parasites using a polyclonal rabbit antibody raised against *T. brucei* luminal binding protein 1 (BiP) (kindly provided by J. Bangs, SUNY, USA) and PAD1 (kindly provided by K. Matthews, University of Edinburgh, UK) to detect stumpy forms, using a Dako Autostainer Link 48 (Dako, Denmark) and were subsequently counterstained with Gill's Haematoxylin. The extent of inflammatory cell infiltration in skin sections was assessed in H&E-stained sections. Stained slides were assessed by two independent pathologists blinded to infection status and experimental procedures. Skin parasite burden was assessed at both intravascular (parasites within the lumen of dermal or subcutaneous small to medium-sized vessels) and extravascular locations (parasites outside blood vessels, scattered in the connective tissue of the dermis or in the subcutaneous adipose tissue) and was evaluated in 5 randomly selected fields at 40X magnification for each sample. A semi-quantitative ordinal score was used to grade the trypanosomes burden in the skin, as follows: 0 = no parasites; 1 = 1 to 19 trypanosomes; 2 = 20–50 trypanosomes; 3 = > 50 trypanosomes. An average parasite burden score per field of view was calculated for each skin section. The severity of skin inflammation was assessed semi-quantitatively, graded on a 0–3 scale: 0 (leukocytes absent or rarely present); 1, mild (low numbers of mixed inflammatory cells present); 2, moderate (increased numbers of mixed inflammatory cells), and 3, marked (extensive numbers/aggregates of inflammatory cells). The average of 10 randomly selected fields at 20X objective magnification for each skin section determined the inflammatory score.

**DNA extraction and quantitative polymerase chain reaction (qPCR) of *T. brucei* in murine skin tissue.** Genomic DNA (gDNA) was extracted from 25–30 mg skin tissue preserved at −80 °C. After defrosting on ice, the skin was finely chopped with scissors, and disrupted for 8 min in 300 μL ATL buffer (Qiagen) using a Qiagen Tissuelyser at 50 Hz with a ceramic bead (MPBio). Disrupted tissues were incubated at 56 °C with 2 mg/ml proteinase K (Invitrogen) overnight and DNA was extracted from digested tissue using the Qiagen DNeasy Blood and Tissue Kit (Qiagen). The resulting gDNA was quantified using a Qubit Fluorimeter (Thermofisher Scientific). and diluted to 4 ng/μl. Trypanosome load in the skin was determined using Taqman real-time PCR, using primers and a probe specifically designed to detect the trypanosome *Pfr2* gene[69]. Reactions were performed in a 25 μl reaction mix comprising 1X Taqman Brilliant III master mix (Agilent, Stockport, UK), 0.2 pmol/μl forward primer (CCAACCGTGTGTTTCCTCCT), 0.2 pmol/μl reverse primer (CGGCAGTAGTTTGACACCTTTTC), 0.1 pmol/μl probe (FAM-CTTGTCTTCTCCTTTTTTGTCTCTTTCCCCCT-TAMRA) (Eurofins Genomics, Germany) and 20 ng template DNA. A standard curve was constructed using a serial 10-fold dilution range: $1 \times 10^6$ to $1 \times 10^1$ copies of PCR 2.1 vector containing the cloned *Pfr2* target sequence (Eurofins Genomics, Germany). The amplification was performed on an ARIAMx system (Agilent, USA) with a thermal profile of 95 °C for 3 min followed by 45 cycles of 95 °C for 5 s and 60 °C for 10 s. The *Pfr2* copy number within each 20 ng DNA skin sample was calculated from the standard curve using the ARIAMx qPCR software (Agilent, USA) as a proxy for

the estimated trypanosome load. Skin biopsies from naïve controls were included to determine the background signal and detection threshold.

**Flow cytometry analysis of skin-dwelling αβ and γδ T lymphocytes.** Murine skin biopsies were harvested and digested as indicated above. The resulting single cell suspensions were resuspended in ice-cold FACS buffer (2 mM EDTA, 5 U/ml DNAse I, 25 mM HEPES and 2.5% Foetal calf serum (FCS) in 1× PBS), blocked with TruStain FcX (anti-mouse CD16/32) antibody (Biolegend, clone 93, Cat # 101320, 1:100), stained with the fixable viability eFluor 780 (eBioscience, Cat # 65-0865-18, 1:1,000 in 1X PBS), and the following anti-mouse antibodies against extracellular surface markers: F4/80-APC-Cy7 (Biolegend, clone BM8, Cat # 1231118, 1:400), CD19-APC-Cy7 (Biolegend, clone 1D3/CD19, Cat # 152412, 1:400), Ter119-APC-Cy7 (Biolegend, clone TER-119, Cat # 116204, 1:400), CD45-PE (Biolegend, clone I3/2,3, Cat # 147712, 1:400), CD3e-PE Dazzle 594 (Biolegend, clone KT3.1.1, Cat # 155620, 1:400), TCRgd-Brilliant Violet 421 (Biolegend, clone GL3, Cat # 118120, 1:400), CD27-APC (Biolegend, clone LG.3A10, Cat # 124212, 1:400), CD45-PE Dazzle 594 (Biolegend, clone 30-F11, Cat # 103146, 1:400), CD4-APC (Biolegend, clone GK1.5, Cat # 100412, 1:400), CD8a-Brilliant Violent 711 (Biolegend, clone 53-6.7, Cat # 100748, 1:400), IFNα-PE (Biolegend, clone XMG1.2, Cat # 505808, 1:400), CD3e-Alexa Fluor 488 (Biolegend, clone 500A2, Cat # 152322, 1:400), IL-17A-PE (Biolegend, clone TC11-18H10.1, Cat # 506904, 1:400). For intracellular staining, single-cell suspensions were stimulated as above in Iscove's modified Dulbecco's media (supplemented with 1X non-essential amino acids, 50 U/ml penicillin, 50 μg/ml streptomycin, 50 μM β-mercaptoethanol, 1 mM sodium pyruvate and 10% FBS (Gibco) containing 1X cell stimulation cocktail containing phorbol 12-myristate 13-acetate (PMA), Ionomycin, and Brefeldin A (eBioSciences™) for 3 h at 37 °C and 5% $CO_2$. Unstimulated controls were also included in these analyses. After surface marker staining, cells were fixed and permeabilized with a FOXP3/Transcription Factor Staining Buffer Set (eBioscience) and stained overnight at 4 °C. The Vγ6 antibody (Pacific blue; 1:50) was kindly provided by Shinya Hatano and Yasunobu Yoshikai (Kyushu University, Fukuoka, Japan). For flow cytometry analysis, samples were run on a flow cytometer LSRFortessa (BD Biosciences) and analysed using FlowJo software version 10 (Treestar).

## Statistics and reproducibility
All statistical analyses were performed using Graph Prism Version 8.0 for Windows or macOS, GraphPad Software (La Jolla California USA). The data distribution was determined by normality testing using the Shapiro-Wilks test. Where indicated, data were analysed by unpaired Student's *t*-test, Mann–Whitney test, or one-way analysis of variance (ANOVA). Data were considered to be significant where $p < 0.05$. For in vivo experiments, we matched the sex and age of the mice in experimental batches using a block design including randomisation of experimental units and repeated at least twice with similar results. The single cell atlas and the spatial transcriptomics experiments were performed once due to financial constraints but were validated in independent experiments using imaging and/or flow cytometry. Data collection and analysis were not performed blindly to the conditions of the experiment due to specific requirements in the animal project license.

## Reporting summary
Further information on research design is available in the Nature Portfolio Reporting Summary linked to this article.

## Data availability
The *Mus musculus* (mm10; https://www.ncbi.nlm.nih.gov/datasets/genome/GCF_000001635.20/) and *Trypanosoma brucei* (TREU927; https://www.ebi.ac.uk/genomes/CH464491.html) references used in

this study can be access using the links provided. The transcriptome data generated in this study have been deposited in the Gene Expression Omnibus GSE226113. The processed transcript count data and cell metadata generated in this study are available at Zenodo (https://doi.org/10.5281/zenodo.7677469; https://zenodo.org/record/7677469). Additional data generated in this study are provided in the Supplementary Information/Source Data file, or in the supplementary data files. Source data are provided with this paper.

## Material availability

## Code availability
Code used to perform analysis described can be accessed at Zenodo (https://doi.org/10.5281/zenodo.7677469).

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

## Acknowledgements

We thank Julie Galbraith and Pawel Herzyk (Glasgow Polyomics, University of Glasgow) for their support with library preparation and sequencing. Similarly, we would like to thank the technical staff at the University of Glasgow Biological Services for their assistance in maintaining optimal husbandry conditions and comfort for the animals used in this study. We would also like to thank Frazer Bell, Lynn Stevenson, and the rest of the University of Glasgow Veterinary school histopathology service for the technical assistance with histological advice and preparations. We thank the Maria Kasper Lab (Karolinska Institutet, Sweden) for their advice on single cell skin dissociation, and Rebecca O'Brien (National Jewish Health, USA) for providing the Vγ4/6−/− mice. We also thank Shinya Hatano and Yasunobu Yoshikai (Kyushu University, Fukuoka, Japan) for providing the Vγ6 antibody. We thank Dr Jean Rodgers for the work conducted under her Home Office Animal License (PPL No. PC8C3B25C). This work was funded by a Sir Henry Wellcome post-doctoral fellowship (221640/Z/20/Z to JFQ), a Wellcome Trust FutureScope grant (104111/Z/14/Z Wellcome Centre for Integrative Parasitology to JFQ), a University of Glasgow Lord Kelvin Adam Smith Leadership Fellowship (to JFQ), and a Wellcome Trust Senior Research fellow (209511/Z/17/Z to AML). RH is a Wellcome Trust PhD student (Wellcome Trust 218518/Z/19/Z). MCS, PC, JO, AC, and NRK are supported by a Wellcome Trust Senior Research fellowship (209511/Z/17/Z) awarded to AML. JO is also supported by a Wellcome Trust FutureScope grant (104111/Z/14/Z Wellcome Centre for Integrative Parasitology to JFQ). SBC is supported by a grant from the Annie McNab Bequest (CRUK Beatson Institute), Breast Cancer Now (2018JulPR1101, 2019DecPhD1349, 2019DecPR1424), Cancer Research Institute (CLIP award), Cancer Research UK (RCCCEA-Nov21\100003, EDDPGM-Nov21\100001, DRCNPG-Jun22\100007), and Pancreatic Cancer Research and Worldwide Cancer Research (22-0135). NAM is supported by a BBSRC Institute Strategic Programme (BBS/E/D/20002173 and BB/X010937/1).

## Author contributions

Conceptualisation: J.F.Q., M.C.S., N.A.M. Methodology: J.F.Q., M.C.S., P.C., R.H., J.O., B.C., A.L., A.C., N.R.K., S.B.C., A.M.L. Formal analysis: J.F.Q., M.C.S., P.C., R.H., J.O., B.C., A.L., N.A.M., D.M.N., N.R.K. Writing – original draft: J.F.Q., M.C.S. Writing – reviewing and editing: J.F.Q., M.C.S., P.C., R.H., J.O., B.C., A.L., A.C., S.B.C., N.R.K., D.M.N., N.A.M., A.M.L. Writing – final edits: J.F.Q. Funding acquisition: J.F.Q., A.M.L.

## Competing interests

The authors declare no competing interests.

## Ethical approval

All animal experiments were approved by the University of Glasgow Ethical Review Committee and performed in accordance with the home office guidelines, UK Animals (Scientific Procedures) Act, 1986 and EU directive 2010/63/EU. All experiments were conducted under SAPO regulations and UK Home Office project licence number PC8C3B25C to Dr. Jean Rodgers.
