## [Peer Review File · Nature Communications]

$\gamma\delta$ T cells control murine subcutaneous adipose wasting during chronic *Trypanosoma brucei* infectionREVIEWER COMMENTS

Reviewer #1 (Remarks to the Author):

Quintana et al. applied single-cell RNA-seq and spatial transcriptomics analyses to study the transcriptomics response of murine skin during *T. brucei* infection. They identified significant changes of cell-type composition, gene expression levels, as well as ligand-receptor activities. This reviewer primarily focuses on the technical aspects of single-cell and spatial transcriptomics analysis, whereas leaving the evaluation of biological significance to other reviewers. The overall data generation and analysis procedure followed standard practice. The data quality seems good. The method description is generally clear. However, there are a number of errors and inaccuracies as listed below.

Specific comments:

1. Line 195-196. Steps 1 and 2. It is unclear why cells with “i) nFeature > 100 or <5,000, ii) nCounts > 100 or <20,000” were filtered out. Is this a typo? It seems to me these cells should be retained for further analysis rather than filtered out. Also, why thresholding the upper bound of nFeature and nCounts? It seems reasonable to have high counts if the cell size is small.
2. Line 215. The authors used Clustree to select the resolution parameter, but it is unclear what the resulting value of resolution is.
3. Line 276-278. NICHES is mentioned in the Methods but not in the Results. It is unclear whether it was used in any analyses presented in the paper. There is no description of the method. A citation is provided Raredon 2022, but the full reference info is missing.
4. Line 434-436. It is unclear how the cell-type annotation was made for the Visium data. Regardless, it seems an over-simplification to use a single cell type to annotate each spot since the data does not have single-cell resolution. Using a published cell-type deconvolution method to estimate the composition of different cell types seems more appropriate.
5. Line 536-539. Module scoring analysis is mentioned multiple times in the paper and presumably is an important component of the analysis. However, there is no description of such analysis in the Methods section.
6. Line 547-550. NicheNet was used for cell-cell communication analysis, but how the procedure was used is unclear. NicheNet first generates prior estimate by integrating information from public databases without using any new data. Is this what the x-label in Figure 4G referring to?

Reviewer #2 (Remarks to the Author):

In this very interesting manuscript, the authors dissected the molecular responses triggered by the skin and fat tissue during an experimental infection with *Trypanosoma brucei*. As new central players in this response the authors identified gamma/delta T cells (mainly Vg6+) which, during a trypanosome infection, increase the levels of IL-17A in skin. Furthermore, in interstitial pre-adipocytes, where parasites can also be found, same cell type is also

activated via Cd40, Il6, Il10, and Tnfsf18 signalling, according to cell-cell- in silico analyses of tissues from infected animals. The authors then analysed the infection responses in IL-17A-producing gamma/delta T cells mice, which showed a dramatic inflammatory response, an increased frequency of skin resident IFN γ -producing CD8 $^+$ T cells and a milder wasting of subcutaneous adipose tissue. Overall, the authors show convincing evidence of the crosstalk between adipocytes and V γ 6 $^+$ cells to control skin inflammation and tissue wasting during an experimental infection with trypanosomes.

I think the paper is very well written, uses state-of-art methodologies and brings novelty in our understanding on how skin inflammation is control during an experimental trypanosome infection (and possibly also by other parasites that colonise the skin of infected mammals). I can envision this work attracting the attention not only of the parasitology and infectious diseases communities, but also from microbiologists, immunologists and physiologists.

Regarding of the importance African trypanosomes have in causing human and animal trypanosomiasis, this work shows a great potential for using trypanosomes as model for understanding fundamental aspects of how skin inflammation gets modulated, including pathogen persistence, especially during the context of a chronic infection. Some of these aspects would be very interesting to investigate when trypanosomes get directly inoculated into the vertebrate's skin after a tsetse infectious bite, as insect saliva on its own modulate the skin's cytokine response and parasites may directly interact with adipocytes after few days (before chronicity is reached). Although this is beyond the scope of this work, it might be worth mentioning it within the Discussion section.

Reviewer #3 (Remarks to the Author):

The MacLeod group pioneered the discovery of African trypanosomes in the skin of undiagnosed, asymptomatic sleeping sickness patients. The current manuscript by Quintana et al reports on an unprecedented single cell and spatial analysis of mouse skin during a *Trypanosoma brucei* infection. Infections in genetic mouse models have explored the role of gd T cells. The research question is very relevant and pertinent as asymptomatic individuals are a constraint on the WHO roadmap goal of gHAT elimination by 2030. Understanding the basis of asymptomatic skin infection is therefore a significant interest. While there are a number of new findings and catalogue of scRNAseq data, several of the presented hypotheses and conclusions stated in the abstract and body text are – although attractive - not supported by the data.

The spatial analyses in Fig. 1E, Fig. 2I and Fig. 4E, F & H are confusing and not convincing as the skin section used for naïve mice seems anatomically quite different from the section used for infected mice. These two sections are used to build all spatial conclusions in the manuscript. In infected mice there is a keratinocyte layer on both sides, a much thicker muscle layer and a thin subcutis. The choice of section makes comparison very difficult, although the H&E stain in Fig. 1A shows that more comparable sections could have been chosen. In the infected mice, CD207 $^+$ LCs are in the epidermis as expected at one side

(bottom) of the section, but seem to have shifted position at the other (top) side (Fig. 1E). As keratinocytes colocalize with the LC, it is not likely that the LCs have truly shifted position from the epidermis to the subcutaneous adipose tissue as stated. This is more likely an artefact of the section. Colocalization of adipose and Vg6+ T cells based on the Tnfsf18 and Tnfrsf18 expression is also not noticeable. In general, the conclusions of subcutaneous adipocytes aiding the stimulation of Vg6+ cells is not apparent from the spatial data.

ScRNAseq was performed on total skin single cell suspensions, resulting in a high proportion of keratinocytes and a small proportion of myeloid/lymphoid immune cells. The used approach resulted in a high variability amongst the two naïve mice. As a result, several of the conclusions drawn in lines 466-472 are not supported by the data. In Fig. 2C, I do not see evidence for significant differences in KC2, KC3, adipocytes, ECs and IPA2 as a result of infection.

Lines 479-482: clustering analysis resulted in the annotation of a cluster as IPA2, contributing to cytokine production (Fig. 2D) and antigen presentation (Fig.2E), but both features seem to be characteristic of different populations within the IPA2 cluster. Can the heterogeneity within this CD34+ progenitor cluster be better defined?

Line 499, Fig. 3B: can the authors comment on CD14+ monocytes as a source of IL-10? Now only Mrc1+ macrophages are mentioned, although Fig. 3B would suggest similar contributions of both.

Lines 509-110, Fig. 3D : can the authors comment on whether they consider mast cells also as important source of (de novo) cytokines?

The increase of Vg6+ T cells in response to infection is clear, but the increase of ILC2 and NK is neither mentioned nor discussed.

The mice used for infection studies are very young, between 6-8 weeks of age. I wonder whether the differences at baseline in subcutaneous adipose area between the FVB/NJ and Vg4/6-/- mice are not just age related. It would be informative to include body weights to demonstrate that differences between groups are not just age/weight-related. It is difficult to make conclusions about infection induced adipose wasting as the deficient mice already have less subcutaneous adipose.

The authors state a role of IL-17A, but do not provide evidence for Vg6+ T cells expressing this cytokine at the transcript or protein level.

It is concluded by the authors that mice deficient in IL-17A-producing gd T cells have an increased frequency of IFNg-producing CD8+ T cells. This is not observed during infection and as such a link between gd T cells, IFNg-producing CD8+ T cells and tissue wasting cannot be substantiated.

Methods:

Lines 143-148: there is some redundancy in the text.

Line 150: hyperlink error in the text

It is not clear what is meant with "all experiments were conducted between 8h and 12h"

Lines 192 and 199: the average reads per cell are the same before and after applying the cut-off which seems incorrect. Can authors include a reference for the parameters and cut-off criteria to filter out the low quality cells?

Line 319: methods describe the use of a polyclonal antibody against BiP, but the PAD1 staining of stumpy forms (Fig. 1) is not described.

Line 374: intracellular Foxp3-staining is described, but the data are not shown or discussed.

Some minor typographical suggestions:

Line 262: *Mus musculus* in italics

Line 298: um -> μm

Line 344: remove the dot

Consistent use of spacing between number and units.

Decision on Nature Communications manuscript NCOMMS-23-09302

We thank all the reviewers for their positive assessments and helpful comments. We have summarised the changes made to the manuscript in response, and we believe they have significantly improved it.

Major points:

1. We have included new, anatomically comparable, skin biopsies in our spatial transcriptomic datasets, as suggested by reviewer 3.
2. We have refined our spatial single cell deconvolution using anchor-based approaches, as suggested by reviewer 1.
3. We have validated several of our *in silico* predictions (e.g., adipocyte-derived *Tnfsf18* as activating factor for T cells, adipocyte- $\gamma\delta$ T cell co-localisation, IL-17 production by $\gamma\delta$ T cells) using smFISH, as suggested by reviewer 3.
4. We have repeated the infections in the V γ 4/6 KO mice to measure bodyweight, as well as the weight of several adipose tissue depots, as suggested by reviewer 3.

Below we have provided point-by-point answers to the major and minor comments raised by these three reviewers.

REVIEWER COMMENTS

Reviewer #1 (Remarks to the Author):

Quintana et al. applied single-cell RNA-seq and spatial transcriptomics analyses to study the transcriptomics response of murine skin during *T. brucei* infection. They identified significant changes of cell-type composition, gene expression levels, as well as ligand-receptor activities. This reviewer primarily focuses on the technical aspects of single-cell and spatial transcriptomics analysis, whereas leaving the evaluation of biological significance to other reviewers. The overall data generation and analysis procedure followed standard practice. The data quality seems good. The method description is generally clear. However, there are a number of errors and inaccuracies as listed below.

1. Line 195-196. Steps 1 and 2. It is unclear why cells with “i) nFeature > 100 or <5,000, ii) nCounts > 100 or <20,000” were filled out. Is this a typo? It seems to me these cells should be retained for further analysis rather than filtered out. Also, why thresholding the upper bound of nFeature and nCounts? It seems reasonable to have high counts if the cell size is small.

Authors: Indeed, this is a typo. We have amended the text as follow:

Line 231 “High quality cells were identified according to the following criteria: i) nFeature > 100 or <5,000, ii) nCounts > 100 or <20,000, iii) > 30% reads mapping to mitochondrial genes, and iv) > 40% reads mapping to ribosomal genes, v) genes detected < 3 cells.”

We typically threshold the upper bound of both nFeature and nCounts to make sure we don't carry over any potential droplets containing 2 or more cells.

2. Line 215. The authors used Clustree to select the resolution parameter, but it is unclear what the resulting value of resolution is.

Authors: We chose a resolution of 0.4 for the global FindClusters function as stated in line 252. We have reworded the text for clarity as follow:

Line 250 “Cluster analysis, marker gene identification, and subclustering. The integrated dataset was then analysed using *RunUMAP* (10 dimensions), followed by *FindNeighbors* (10 dimensions, reduction = “pca”) and *FindClusters* (resolution = 0.4). The resolution used for these analyses was selected using the Clustree package²⁶ (Supplementary Figure 1). With this approach, we identified a total of 16 cell clusters”

3. Line 276-278. NICHES is mentioned in the Methods but not in the Results. It is unclear whether it was used in any analyses presented in the paper. There is no description of the method. A citation is provided Raredon 2022, but the full reference info is missing.

Authors: We apologise for this error. We used NicheNet for cell-cell interaction predictions from our single cell dataset. We have now removed any reference from NICHES from the methods section in line 287 – 289.

4. Line 434-436. It is unclear how the cell-type annotation was made for the Visium data. Regardless, it seems an over-simplification to use a single cell type to annotate each spot since the data does not have single-cell resolution. Using a published cell-type deconvolution method to estimate the composition of different cell types seems more appropriate.

Authors: We thank this reviewer for this useful comment. We have now tried several publicly available packages for data integration, including anchor-based approaches (Seurat) as well as other integration packages (e.g., Cell2Location, Giotto). The most compelling results from data integration were obtained with anchor-based integration approaches using Seurat, as recently shown (Castillo et al., *Sci. Immunol.* 8, eabq7991, 2023). The results have now been incorporated on figures 1F and 3C.

We have also amended the text to reflect this as follows:

Line 344 “For single cell and spatial transcriptomics data integration, we used the anchor-based integration workflow as described in the Seurat package vignette. Briefly, after quality control and filtering, spatial transcriptomics samples were normalised independently employing variance stabilisation through the *SCTransform* function with default parameters. We then identified anchors to be transferred between datasets using the function *FindTransferAnchors* and *TransferData* with default parameters and a total of 30 dimensions”

Line 576 “In the spatial context, the predicted distribution of the various stromal cell populations identified in our single cell atlas was heterogeneous. For instance, the keratinocyte and fibroblast clusters were predicted to be predominantly found in the epidermal layers of the skin biopsies, whereas the melanocytes and adipocytes were preferentially predicted to be in the dermis and hypodermis in both skin biopsies examined (**Figure 1F**). We also noted that Langerhans cells and myeloid cells (containing macrophages and dendritic cells) were preferentially found in the epidermis of the skin of naïve animals, perhaps around the perifollicular spaces as recently found in human skin³¹, but also localised to the dermal and hypodermal regions during infection, coinciding with the localisation of the adipose tissue (**Figure 1F**).

5. Line 536-539. Module scoring analysis is mentioned multiple times in the paper and presumably is an important component of the analysis. However, there is no description of such analysis in the Methods section.

Authors: We apologise for the omission and have added the following text in the methods section:

Line 277 “Module scoring were calculated using the *AddModuleScore* function to assign scores to groups of genes of interest (*Ctrl* = 100, *seed* = NULL, *pool* = NULL), and the scores were then represented in feature plots. This tool measures the average expression levels of a set of genes, subtracted by the average expression of randomly selected control genes.”

6. Line 547-550. NicheNet was used for cell-cell communication analysis, but how the procedure was used is unclear. NicheNet first generates prior estimate by integrating information from public databases without using any new data. Is this what the x-label in Figure 4G referring to?

Authors: Indeed, the heatmap is colour-coded based on the probability of ligand-receptor pair interactions estimated by NicheNet. Please note that the bottom part of the graph only refers to the predicted strength of the predicted interaction potential (colour coded) and not to a label on the X axis; The stronger the colour the higher probability of the interaction to be real.

Reviewer #2 (Remarks to the Author):

In this very interesting manuscript, the authors dissected the molecular responses triggered by the skin and fat tissue during an experimental infection with *Trypanosoma brucei*. As new central players in this response the authors identified gamma/delta T cells (mainly Vg6+) which, during a trypanosome infection, increase the levels of IL-17A in skin. Furthermore, in interstitial pre-adipocytes, where parasites can also be found, same cell type is also activated via Cd40, Il6, Il10, and Tnfsf18 signalling, according to cell-cell- in silico analyses of tissues from infected animals. The authors then analysed the infection responses in IL-17A-producing gamma/delta T cells mice, which showed a dramatic inflammatory response, an increased frequency of skin resident IFNg-producing CD8+ T cells and a milder wasting of subcutaneous adipose tissue. Overall, the authors show convincing evidence of the crosstalk between adipocytes and Vg6+ cells to control skin inflammation and tissue wasting during an experimental infection with trypanosomes.

I think the paper is very well written, uses state-of-art methodologies and brings novelty in our understanding on how skin inflammation is controlled during an experimental trypanosome infection (and possibly also by other parasites that colonise the skin of infected mammals). I can envision this work attracting the attention not only of the parasitology and infectious diseases communities, but also from microbiologists, immunologists and physiologists.

Regarding of the importance African trypanosomes have in causing human and animal trypanosomiasis, this work shows a great potential for using trypanosomes as model for understanding fundamental aspects of how skin inflammation gets modulated, including pathogen persistence, especially during the context of a chronic infection. Some of these aspects would be very interesting to investigate when trypanosomes get directly inoculated into the vertebrate's skin after a tsetse infectious bite, as insect saliva on its own modulate the skin's cytokine response and parasites may directly interact with adipocytes after few days (before chronicity is reached). Although this is beyond the scope of this work, it might be worth mentioning it within the Discussion section.

Authors: We sincerely thank this reviewer for their positive feedback. We have included a comment regarding natural infections in the discussion section as follow:

Line 1187 “Thus, in the context of chronic skin infection with *T. brucei*, we propose a model whereby subcutaneous adipocytes (in addition to Langerhans cells and keratinocytes) have a critical role as coordinators of local innate and adaptive immune responses. In the context of trypanosome infection, subcutaneous adipocytes may detect the presence of parasites (e.g., via Toll-like receptor signalling) to trigger the recruitment and activation of innate immune cells such as $\gamma\delta$ T cells to mobilise energy stores to meet the energetic requirements needed to control infection, as recently proposed⁶⁰. In this manuscript, we focus on characterising the immunological events

that the parasites encounter in the skin prior to forward transmission (host to vector). However, given the immunomodulatory nature of the salivary components delivered at the site of inoculation by tsetse flies^{61–64}, it would also be interesting to explore the dynamics of $\gamma\delta$ T cell activation and adipocyte responses in the skin during the onset of infection.

Reviewer #3 (Remarks to the Author):

The MacLeod group pioneered the discovery of African trypanosomes in the skin of undiagnosed, asymptomatic sleeping sickness patients. The current manuscript by Quintana et al reports on an unprecedented single cell and spatial analysis of mouse skin during a *Trypanosoma brucei* infection. Infections in genetic mouse models have explored the role of gd T cells. The research question is very relevant and pertinent as asymptomatic individuals are a constraint on the WHO roadmap goal of gHAT elimination by 2030. Understanding the basis of asymptomatic skin infection is therefore a significant interest. While there are a number of new findings and catalogue of scRNAseq data, several of the presented hypotheses and conclusions stated in the abstract and body text are – although attractive - not supported by the data.

Authors: We thank this reviewer for such thorough and detailed feedback. We have provided additional data to support our conclusions, which we believe makes a stronger case and improves the manuscript.

The spatial analyses in Fig. 1E, Fig. 2I and Fig. 4E, F & H are confusing and not convincing as the skin section used for naïve mice seems anatomically quite different from the section used for infected mice. These two sections are used to build all spatial conclusions in the manuscript. In infected mice there is a keratinocyte layer on both sides, a much thicker muscle layer and a thin subcutis. The choice of section makes comparison very difficult, although the H&E stain in Fig. 1A shows that more comparable sections could have been chosen. In the infected mice, CD207+ LCs are in the epidermis as expected at one side (bottom) of the section, but seem to have shifted position at the other (top) side (Fig. 1E). As keratinocytes colocalize with the LC, it is not likely that the LCs have truly shifted position from the epidermis to the subcutaneous adipose tissue as stated. This is more likely an artefact of the section. Colocalization of adipose and Vg6+ T cells based on the *Tnfsf18* and *Tnfrsf18* expression is also not noticeable. In general, the conclusions of subcutaneous adipocytes aiding the stimulation of Vg6+ cells is not apparent from the spatial data.

Authors: We have replaced the previous infected skin spatial transcriptomics dataset with a new dataset from an unfolded skin section, which is overall comparable to the naïve section. We have also included smFISH analysis of infected skin sections to validate the *in silico* predictions generated from our combined single cell and spatial datasets. In particular, we have included *Cd207* for Langerhans' cells (**Figure 3D**) and *Tcrg-V6* for $V\gamma6^+$ T cells (**Figure 4F**). Similarly, we have included an additional imaging experiment demonstrating the spatial distribution of $V\gamma6^+$ T cells in proximity to *Tnfsf18*⁺ adipocytes during infection (**Figure 4I**), which we believe support our prediction of a potential adipocyte- $\gamma\delta$ T cell interaction. We have added the following text in the manuscript to reflect these changes:

Line 803 “Furthermore, the *Cd207*⁺ LCs were predicted to be predominantly in the epidermis/dermis of naïve animals but were also found within the adipose tissue in skin sections from infected animals, coinciding with adipocytes and several populations of progenitor keratinocyte populations (**Figure 1E, 3C, and S3 Figure**). The spatial distribution of LCs in skin biopsies from naïve and infected animals was independently validated using smFISH, which confirms the accumulation of *Cd207*⁺ LCs and *Cd4*⁺ T cells around the perifollicular spaces in the epidermis/dermis in naïve samples, but were also detected in the adipose tissue within the hypodermis during infection (**Figure 3D**), consistent with previous studies demonstrating that CD207⁺ LCs display a migratory behaviour during infection³⁴”

Line 875 “Indeed, we found *V γ 6*⁺ cells in the dermis and epidermis, as well as in the hypodermis of skin biopsies from naïve and infected animals from independent tissue sections using smFISH, validating our *in silico* predictions (**Figure 4I**)”

Line 924 “We validated this pattern of expression using smFISH and found that *Tnfrsf18* was upregulated in the parenchyma of the subcutaneous adipose tissue, coinciding with *Adipoq*⁺ adipocytes and *Trcd*⁺ $\gamma\delta$ T cells (**Figure 4I**).”

ScRNAseq was performed on total skin single cell suspensions, resulting in a high proportion of keratinocytes and a small proportion of myeloid/lymphoid immune cells. The used approach resulted in a high variability amongst the two naïve mice. As a result, several of the conclusions drawn in lines 466-472 are not supported by the data. In Fig. 2C, I do not see evidence for significant differences in KC2, KC3, adipocytes, ECs and IPA2 as a result of infection.

Authors: The ratios between CD45⁻ and CD45⁺ retrieved from our single cell data are consistent with previous reports in which the majority of the skin-derived cells are either keratinocytes or fibroblasts (Castillo et al., *Sci. Immunol.* 8, eabq7991, 2023). We agree that the overall cell frequency is variable in the naïve samples, and thus we have removed the comparisons regarding the KC subpopulations from the text. However, we have highlighted a higher frequency of both IPAs and endothelial cells in the text (and figure 2C, right panel), as follow:

Line 613 “These two populations of *Dpp4*⁺ *Pi16*⁺ interstitial preadipocytes (IPAs), *Thy1*⁺ (IPA1) and *Cd34*⁺ *Ly6a*⁺ (IPA2), likely represent cells at different developmental stages within the adipocyte trajectory. Some stromal populations were altered during infection. For instance, we observed a higher frequency of endothelial cells (ECs; from 6.67% to 8.02% in naïve and infected, respectively), IPA1 (from 4.01% to 5.81% in naïve and infected, respectively), and IPA2 (from 1.98% to 3.81% in naïve and infected, respectively) (**Figure 2C**)”

Lines 479-482: clustering analysis resulted in the annotation of a cluster as IPA2, contributing to cytokine production (Fig. 2D) and antigen presentation (Fig.2E), but both features seem to be characteristic of different populations within the IPA2 cluster. Can the heterogeneity within this CD34⁺ progenitor cluster be better defined?

Authors: It is indeed possible that IPA2 encompass progenitor cells at different stages of differentiation contributing to the overall heterogeneity in the expression

levels reported within this cluster. As shown in Figure 2B, IPA2 has a higher expression level of *Cd34* and *Ly6a*, a marker of precursor cells, compared to IPA1, which might reflect different developmental stages within the interstitial preadipocyte population. To our knowledge, the immunological competence of interstitial preadipocytes has not been reported, even less so in the context of skin infections.

It is important to stress that we are unable to offer additional resolution within the IPA2 cluster of this dataset. Also, a further characterisation of the diversity of the subcutaneous adipose tissue IPAs falls out of the scope of this manuscript. However, we are currently investigating this in more detail in a separate manuscript (DOI: <https://www.biorxiv.org/content/10.1101/2022.09.23.509158v3>).

We have included the following text in the manuscript to clarify further:

Line 613 “These two populations of *Dpp4⁺ Pi16⁺* interstitial preadipocytes (IPAs), *Thy1⁺* (IPA1) and *Cd34⁺ Ly6a⁺* (IPA2), likely represent cells at different developmental stages within the adipocyte trajectory.”

Line 499, Fig. 3B: can the authors comment on CD14⁺ monocytes as a source of IL-10? Now only *Mrc1⁺* macrophages are mentioned, although Fig. 3B would suggest similar contributions of both.

Authors: Indeed, both clusters express *Il10*. We have amended the text as follow:

Line 792 “Interestingly, in addition to *Cd14⁺* monocytes, *Mrc1⁺* macrophages also expressed *Il10*, suggesting that these cells may have anti-inflammatory properties (Figure 3B).”

Lines 509-110, Fig. 3D: can the authors comment on whether they consider mast cells also as important source of (de novo) cytokines? The increase of *Vg6⁺* T cells in response to infection is clear, but the increase of ILC2 and NK is neither mentioned nor discussed.

Authors: Indeed, it is likely that other immune cells found in the infected skin contribute to the pool of cytokines secreted locally in response to infection. We have amended the text as follow to reflect this:

Line 819 “Other immune cells within this cluster, in particular mast cells, are also likely to contribute to the pool of cytokines produced locally. For instance, the mast cell cluster express high levels of *Il4*, *Il13*, and *Csf1* (S1D Table), suggesting that these cells engage with Th2 responses to promote tissue repair.”

Line 835 “Overall, we noted an expansion within the T cell compartment in response to infection. For instance, we noted a 4.2-fold increase in the frequency of *Ncr1⁺* NK cells (5.2% vs 21.9% in naïve and infected skin, respectively), a 1.35-fold increase in the frequency of ILC2s (9.86% vs 13.37% in naïve and infected skin, respectively) (Figure 4A). Intriguingly, we noted that the *Vγ6⁺* cell cluster 1 was only present in the naïve skin but reduced upon infection (40.38% vs 5.09% in naïve and infected skin, respectively) (Figure 4A). In contrast, there was a 92-fold increase in the frequency of cells within *Vg6⁺* cluster 2 cells in response to infection (0.22% vs

20.88% in naïve and infected skin, respectively) (**Figure 4A**). *In vivo*, we observed a significant expansion of CD27⁻ (IL-17A-producing) $\gamma\delta$ T cells and a concomitant reduction in the frequency of CD27⁺ (IFN γ -producing) $\gamma\delta$ T cells in skin biopsies from infected BALB/c mice compared to naïve controls (**Figure 4C**), following the same trend predicted by the scRNAseq data (**Figure 4A and 4B**), thus validating our *in silico* prediction.”

Line 874 “Together, these results indicate that during infection there is an expansion of *Nrc1*⁺ NK cells and ILC2s, that are likely to contribute to the cytokine-mediated inflammation observed in the skin during infection.”

The mice used for infection studies are very young, between 6-8 weeks of age. I wonder whether the differences at baseline in subcutaneous adipose area between the FVB/NJ and *V γ 4/6^{-/-}* mice are not just age related. It would be informative to include body weights to demonstrate that differences between groups are not just age/weight-related. It is difficult to make conclusions about infection induced adipose wasting as the deficient mice already have less subcutaneous adipose.

Authors: All the mice used in the comparison between FVB/NJ and *V γ 4/6^{-/-}* were of the same age, to control for this potential caveat. As this reviewer rightly points out, the *V γ 4/6^{-/-}* mice appear to have less subcutaneous adipose tissue compared to FVB/NJ mice, and we believe this is associated with a direct effect of local IL-17A (likely produced by *V γ 6⁺* T cells under normal conditions) supporting adipose tissue homeostasis. We have made some comments on this regard on a separate manuscript (DOI: <https://www.biorxiv.org/content/10.1101/2022.09.23.509158v3>) and are currently investigating this in more detail. Although we agree that there might exist an age effect on adipose tissue accumulation/dynamics in the *V γ 4/6^{-/-}* mice, this falls out of the scope of this study, but we are currently investigating this in more detail. It is important to stress that the comparisons presented in figure 5 are made against healthy, uninfected mice from the same genetic background to avoid differences such as the ones highlighted by this reviewer confounding our results.

As recommended by this reviewer, we have repeated the infections in both FVB/NJ and *V γ 4/6^{-/-}* mice to measure body weight over the course of infection, as well as the weight of the gonadal white adipose tissue and spleen normalised to body weight (**Figure 5C, D, and Supplementary figure 5B**). Consistent with our morphometric analysis, we did not detect changes in the subcutaneous adipose tissue mass in the *V γ 4/6^{-/-}* mice but the other depots and the spleen did exhibit significant changes. We interpreted these results as a potential site-specific response, maybe attributed to functional differences between the subcutaneous and the gonadal adipose tissue.

Line 961 “Interestingly, there were not significant differences in the bodyweight of FVB/NJ and *V γ 4/6^{-/-}* mice over the course of infection (**Figure 5C**), but we noted that the mass of the subcutaneous adipose tissue (normalised to bodyweight) was significantly reduced in the FVB/NJ mice but not in the *V γ 4/6^{-/-}* mice (**Figure 5D**), indicating that *V γ 4⁺* and *V γ 6⁺* cells are involved in subcutaneous adipose tissue wasting. Furthermore, these effects seem to be restricted to the subcutaneous adipose tissue as we failed to detect significant differences in the spleen and gonadal white adipose tissue mass in response to infection between strains (**Figure S5B**).

To further understand the impact of $V\gamma 4^+$ and $V\gamma 6^+$ cells on adipose tissue responses to infection, we next examined histological sections of skin samples from infected $V\gamma 4/6^{-/-}$ and FVB/N mice, as well as their counterpart naïve controls. Compared to infected FVB/N mice, the $V\gamma 4/6^{-/-}$ mice displayed more severe signs of skin inflammation. Specifically, we observed higher follicular atrophy in the dermis and hypodermis, as well as diffuse lymphocyte aggregates containing large number of plasma cells and oedema in the subcutaneous adipose tissue compared to infected FVB/N mice (**Figure 5E and S3 Table**). Histological analysis indicated that, during infection, FVB/N mice lose a greater proportion of subcutaneous adipocytes than their naïve counterparts, consistent with previous reports in the trypanotolerant C57BL/6 background²⁴ (**Figure 5E and 5F**). In contrast, infected $V\gamma 4/6^{-/-}$ mice retain similar adipocyte numbers to their naïve counterparts (**Figure 5E-5G**), highlighting a critical role for $V\gamma 4/6$ $\gamma\delta$ T cells in modulating subcutaneous adipose tissue wasting. Morphometric analysis revealed that adipocytes in the $V\gamma 4/6^{-/-}$ mice were significantly smaller in area in naïve animals compared to the FVB/N background (**Figure 5E-5G**), potentially highlighting a role for the $V\gamma 4^+$ and $V\gamma 6^+$ cells in maintaining adipocyte function under homeostasis. Moreover, our morphometric analyses revealed that adipocytes within the subcutaneous adipose tissue of infected FVB/N mice were significantly smaller than those in naïve mice, whereas infection did not significantly impact adipocyte size in the $V\gamma 4/6^{-/-}$ mice (**Figure 5E-5G**)."

Line 1157 "This was limited to the subcutaneous white adipose tissue, as the gonadal white adipose tissue was equally wasted in both wildtype and $V\gamma 4/6^{-/-}$ mice, potentially arguing in favour of potential functional differences between white adipose tissue depots"

The authors state a role of IL-17A, but do not provide evidence for $V\gamma 6^+$ T cells expressing this cytokine at the transcript or protein level.

Authors: Work from Sarina Ravens' laboratory and Seth Coffelt laboratory recently demonstrated that $CD27^- V\gamma 6^+ \gamma\delta$ T cells are natural IL-17A producers (Tan L, et al. Cell reports, 27(12): p3657-3671, DOI: 10.1016/j.celrep.2019.05.064; Edwards SC, et al. JEM, 220(2): e20211431). The transcriptional profile of the $V\gamma 6^+$ T cells identified in our skin single cell atlas share many features of those reported previously in these studies, suggesting that these are indeed IL-17 producing $CD27^- V\gamma 6^+ \gamma\delta$ T cells. However, to confirm this we have included validation using flow cytometry showing an expansion of $IL-17A^+ V\gamma 6^+ \gamma\delta$ T cells *in vivo* in response to chronic *T. brucei* infection (**Supplementary Figure 3**). We have also included the following text in the manuscript to reflect this:

Line 854 "Additionally, we detected a significant expansion in the population of $IL-17^+ V\gamma 6^+$ cells in the skin of infected mice compared to naïve controls (**S3A and S3B Figure**), where $V\gamma 6^+$ represent approximate 40% of all the dermal $\gamma\delta$ T cells (**S3A and S3B Figure**), confirming our *in silico* predictions."

It is concluded by the authors that mice deficient in IL-17A-producing $\gamma\delta$ T cells have an increased frequency of IFN γ -producing $CD8^+$ T cells. This is not observed

during infection and as such a link between gd T cells, IFN γ -producing CD8 $^+$ T cells and tissue wasting cannot be substantiated.

Authors: We apologise for this confusion. We did not intent to imply that IFN γ^+ CD8 $^+$ T cells were involved in driving adipose tissue wasting. Rather, we intended to understand why the skin of the V γ 4/6 $^{-/-}$ mice was more inflamed than the wildtype mice, as observed by histological scoring in table S3. Given the important of T cells in mediating inflammatory responses in the skin in the context of other pathologies, we decided to explore this immune subset in more detail. Unexpectedly, we found that the skin of V γ 4/6 $^{-/-}$ mice have a heightened frequency of IFN γ^+ CD8 T cells, which might suggest that V γ 4/6 are involved (directly or indirectly) in restraining IFN γ production by CD8 $^+$ T cells under homeostasis and disease. This could potentially explain why the skin of the V γ 4/6 $^{-/-}$ mice show greater signs of inflammation compared to wild type controls and thus we considered this to be relevant in the story presented here. However, the link of this potentially exacerbated CD8 $^+$ T cell responses to adipose tissue homeostasis and wasting remains to be explored. We have revised the text where appropriate to improve clarity on this point in the results and discussion sections, as follow:

Line 130 “We conclude that IL-17A-producing V γ 6 $^+$ cells are critical mediators of skin immunity against *T. brucei* infection, likely acting both on restraining IFN γ -mediated CD8 $^+$ T cell responses in the skin, and promoting subcutaneous adipose tissue wasting, supporting our recently identified role of IL-17 signalling as a mediator of adipose tissue wasting during *T. brucei* infection²⁴.”

Line 1005 “As expected, within the CD8 $^+$ T cell compartment, we observed that *T. brucei* infection induced a significant increase in the frequency of activated dermal CD8 $^+$ T cells prone to produce IFN γ when challenged *ex vivo* compared to naïve controls (**S6 Figure**). In the skin of the V γ 4/6 $^{-/-}$ mice, there was a significant increase in the frequency of activated IFN γ -producing CD8 $^+$ T cells compared to FVB/N controls under homeostatic conditions, which were exacerbated during infection, and coinciding with the increased pathology scoring of the skin from V γ 4/6 $^{-/-}$ mice. Altogether, these observations suggest that V γ 4 $^+$ and/or V γ 6 $^+$ cells are important to restrain homeostatic and inflammatory CD8 $^+$ T cells responses in the skin, either directly (e.g., controlling the activation threshold) or indirectly (e.g., through the recruitment/activation of other partners involve in CD8 T cell activation)”

Line 1142 “We found that the increased inflammation in the skin of infected V γ 4/6 $^{-/-}$ mice may be mediated, at least in part, by controlling the activation threshold of skin-resident CD8 $^+$ T cells. The increased capacity of CD8 $^+$ T cells to produce IFN γ in the skin of naïve V γ 4/6 $^{-/-}$ mice suggests that these cells play a role in constraining CD8 $^+$ T cell activity under homeostasis. Indeed, the potential capacity of dermal $\gamma\delta$ T cells (including V γ 6 $^+$ cells) to limit pathogenic CD8 $^+$ T responses might be associated with expression of immunomodulatory mediators such as PD-1 (*Pdcd1*)⁵⁸ and Galectin-1 (*Lgals1*)⁵⁹, the latter being highly expressed in the cells within V γ 6 $^+$ cells cluster 1 in our dataset. An alternative possibility is that the increased severity of skin inflammation in infected V γ 4/6 $^{-/-}$ mice is due to exacerbated recruitment of neutrophils, as reported in metastatic breast cancer⁶⁰. These hypotheses remain to

be explored in more detail in future studies.”

Methods:

Lines 143-148: there is some redundancy in the text.

Authors: We have amended the text to resolve this redundancy.

Line 150: hyperlink error in the text

Authors: We have removed this from the text.

It is not clear what is meant with “all experiments were conducted between 8h and 12h”

Authors: All the experiments were conducted in the morning, between 8am and noon. We have amended the text as follow:

Line 179 “All the experiments were conducted in the morning, between 8am and noon”

Lines 192 and 199: the average reads per cell are the same before and after applying the cut-off which seems incorrect. Can authors include a reference for the parameters and cut-off criteria to filter out the low quality cells?

Authors: The parameters to remove low quality cells are described in **line 231**. The overall number of reads did not change after filtering “low quality” cells but we detected less genes per cell.

Line 319: methods describe the use of a polyclonal antibody against BiP, but the PAD1 staining of stumpy forms (Fig. 1) is not described.

Authors: We have amended the text as follow:

Line 404 “Paraffin-embedded skin samples were cut into 3 μm sections and stained for *T. brucei* parasites using a polyclonal rabbit antibody raised against *T. brucei* luminal binding protein 1 (BiP) (J. Bangs, SUNY, USA) and PAD1 (K. Matthews, University of Edinburgh, UK) to detect stumpy forms, using a Dako Autostainer Link 48 (Dako, Denmark) and were subsequently counterstained with Gill’s Haematoxylin.”

Line 374: intracellular Foxp3-staining is described, but the data are not shown or discussed.

Authors: We did not stain for Foxp3. We used an intracellular permeabilization kit called Foxp3/Transcription Factor Staining buffer set from eBioscience, as stated in **line 473**.

Some minor typographical suggestions:

Line 262: *Mus musculus* in italics

Authors: We have corrected this.

Line 298: um -> μm

Authors: We have corrected this.

Line 344: remove the dot

Authors: We have corrected this.

Consistent use of spacing between number and units.

Authors: We have corrected this.

REVIEWER COMMENTS

Reviewer #1 (Remarks to the Author):

The revised manuscript has largely addressed my previous concerns. I just have a few additional comments, which require additional analysis.

1. the authors tried to eliminate doublets by setting an upper bound for nFeature and nCounts. While this is fine as an approximation, it seems over-simplistic. There are established methods that can address this issue more systematically, such as scrublet and DoubletFinder.
2. My comment about cell-type annotation in Visium data remains unresolved. Each spot contains multiple cells typically from different cell types. Therefore it cannot be simply represented by a single cell type. The Seurat analysis employed in the revision does not address this issue, since it attempts to annotate cell types based on the best matching single cell.

Reviewer #2 (Remarks to the Author):

The authors have significantly improved the manuscript. The four major aspects addressed in the point-by-point response letter, including FISH validation of the in silico predictions, have made the experimental evidence and the whole story much stronger. I have no more requests to make and would like to congratulate the authors for this important contribution, which goes beyond the parasitology and host-parasite interactions field.

Reviewer #3 (Remarks to the Author):

The authors have thoroughly addressed all comments raised and provided additional experimental data to support the conclusions drawn.

I only have two minor suggestions for the authors' consideration, purely to help the readers with interpretation of the figures:

- Fig 2C: the order of the cell types with the colour code is discontinuous (IPA1 and 2)
- Fig 4I: label panels as naïve and infected as in Fig. 3D and 4F

Decision on Nature Communications manuscript NCOMMS-23-09302A

We thank all the reviewers for their second round of comments. We hope that these additional revisions satisfy the remaining concerns. Below we have provided point-by-point answers to the remaining comments raised by reviewers 1 and 3 (in blue).

Reviewer #1 (Remarks to the Author):

Authors: We thank this reviewer for their time in providing additional feedback. We hope that these additional analyses satisfy the remaining concerns raised by this reviewer.

The revised manuscript has largely addressed my previous concerns. I just have a few additional comments, which require additional analysis.

1. the authors tried to eliminate doublets by setting an upper bound for nFeature and nCounts. While this is fine as an approximation, it seems over-simplistic. There are established methods that can address this issue more systematically, such as scrublet and DoubletFinder.

Authors: As stated by this reviewer, implementing nFeatures and nCounts upper boundaries as cut-off have been widely used in the literature to generate *in silico* predictions and testable hypotheses from single cell dataset. However, we have repeated our analysis once more, including a DoubletFinder analysis step. After applying the DoubletFinder (DF) pipeline (pN = 0.0675; pK = 0.01 – both parameters optimised using the *paramSweep* function), we found < 4% of the cells in our dataset are predicted to be doublets.

Sample	No. Cells before DF	No. Cells retained After DF	% Predicted doublets
Naïve 1	5,716	5,498	3.81%
Naïve 2	20,084	19,784	1.49%
Infected 1	21,184	20,765	1.98%
Infected 2	9,892	9,498	3.98%

Furthermore, removing the doublets identified by the DoubletFinder package prior to downstream analysis did not render different results to the ones obtained with the nCounts/nFeatures upper boundary cut-off approach. In both cases, we identified 16 clusters encompassing *Fabp5*⁺ keratinocytes, *Col1a1*⁺ fibroblasts, *Cldn5*⁺ endothelial cells, *Pparg*⁺ adipocytes, *Mlana*⁺ melanocytes, *Lyz2*⁺ myeloid cells, and *Cd3g*⁺ T cells, as shown below. Thus, although seemingly simplistic, we consider that the approach we used for the analysis reported in this manuscript using faithfully recapitulates the results obtained with DoubletFinder and can be regarded as comparable.

We have included the DoubletFinder results in supplementary figure 1C and S1A Table, as well as in the methods sections to clarify, as follow:

Line 236 “The DoubletFinder packaged²⁶ (pN = 0.0675, pK = 0.01, using the *paramSweep* function to identified optimal values) identified <4% of predicted doublets (**S1C Figure and S1A Table**).”

Line 1384 (C) Dimensionality plot depicting the number of predicted singlets and doublets using DoubletFinder. KC: keratinocytes, FB: fibroblasts, EC: endothelial cells, MCs: Macrophages, LC: Langerhans cells, Adipo: adipocytes, Erythro: erythrocytes.”

2. My comment about cell-type annotation in Visium data remains unresolved. Each spot contains multiple cells typically from different cell types. Therefore it cannot be simply represented by a single cell type. The Seurat analysis employed in the revision does not address this issue, since it attempts to annotate cell types based on the best matching single cell.

Authors: In our previous point-by-point document, we explained the reasons behind choosing Seurat as a way to deconvolve our spatial transcriptomics data. However, we have now

included additional analysis using the Giotto package in **supplementary figure 2B** and have amended the text and figure captions as stated below. It is important to stress that we were unable to conduct meaningful spatial deconvolution analyses using the specific subsets as inputs (e.g., T cells, non-immune cells such as keratinocytes, etc). In short, when using the individual subsets as input (e.g., myeloid cells, T cells), Giotto renders biologically implausible outputs. For instance, as shown below, when we used the T cell subset as input, Giotto predicted that the epidermis is composed exclusively of $V\gamma6^+$ T cells and $CD4^+$ T cells. Although $V\gamma6^+$ T cells are indeed found in the epidermis, as shown in Figure 4E and 4F, there are other cell types in these tissue areas, indicating that the prediction is not entirely correct when limiting the spot deconvolution to the various subsets reported in our manuscript. Therefore, we decided to maintain our spatial projections using Seurat for the specific subsets, which have all been validated by independent approaches in independent experiments, while also incorporating the Giotto results in Supplementary Figure 2B.

A more thorough comparison of the outputs obtained with different spatial deconvolution packages, or the computational reasons why Giotto fails to consider “true” spot diversity when using the subsets as inputs (e.g., when only a fraction of the reads explain the spot-level transcriptome) fall out of the scope of this manuscript, which has already been supported by extensive *in vivo* experiments supporting the *in silico* predictions generated from our combined approach. We hope that this reviewer is now satisfied with the additional supplementary figure and text, as shown below:

Line 347 “For single cell and spatial transcriptomics data integration, we used the anchor-based integration workflow as described in the Seurat package vignette and the Giotto package²⁷, with default parameters (30 dimensions, resolution = 0.4, k = 15, n_iterations = 1000). Briefly, after quality control and filtering, spatial transcriptomic samples were normalised independently, employing variance stabilisation through the *SCTransform* function with default parameters. We then identified anchors to be transferred between datasets using the function *FindTransferAnchors* and *TransferData* with default parameters and a total of 30 dimensions.”

Line 578 “In the spatial context, the predicted distribution of the various stromal cell populations identified in our single cell atlas was heterogeneous (**Figure 1F, 1G, and Supplementary Figure 2A**). For instance, the keratinocyte and fibroblast clusters were predicted to be predominantly found in the epidermal layers of the skin biopsies, in particular keratinocytes clusters 2 and 3 (**Supplementary Figure 2B**), whereas the melanocytes and adipocytes, and discrete keratinocyte clusters (clusters 4, 7, and 8) were preferentially predicted to be in the dermis and hypodermis in both skin biopsies examined (**Figure 1F and S2B Figure**). We also noted that Langerhans and myeloid cells (containing macrophages and dendritic cells) were preferentially found in the epidermis of the skin of naïve animals, around the perifollicular spaces as recently found in human skin³³. During infection, Langerhans and myeloid cells also localised to the dermal and hypodermal regions, localising with the adipose

tissue (**Figure 1F and S2B Figure**). The skin immune compartment consisted of *Cd3g⁺ Trdc⁺* T cells (1,103 cells), *Cd207⁺* Langerhans cells (LCs; 854 cells), *Lyz2⁺* myeloid cells (1,499 cells), and erythrocytes (479 cells) (**Figure 1C and 1D**). The majority of these immune cells were lowly detected in naïve skin but were readily found in the subcutaneous adipose tissue of infected samples (**Figure 1G and S2B Figure**). Cells within the myeloid (including Langerhans cells) and Lymphoid compartment (T cells) were identified mostly in the epidermis in the naïve skin section (**Figure 1G and S2B Figure**), but were also predicted to occupy niches within the adipose tissue layer in both naïve and infected samples (**Figure 1G and S2B Figure**). Together, these data predict a local distribution of immune and stromal cells within the subcutaneous adipose tissue layer in response to *T. brucei* infection.”

Line 1389 “**Supplementary figure 2. Quality control of 10X Visium datasets from the mouse skin over the course of infection with *T. brucei*.** **A)** Spatial clusters and marker genes for each spatial cluster in the naïve (**left panel**) and infected (**right panel**) murine skin using 10X Visium spatial transcriptomics. **B)** Spatial transcriptomics spot deconvolution analysis for the naïve (top) and infected (bottom) skin sections using the Giotto package. The spot deconvolution was conducted using the single cell data in Figure 1C. Scale bar, 100 mm. KC: keratinocytes, FB: fibroblasts, EC: endothelial cells, MCs: Macrophages, LC: Langerhans cells, Adipo: adipocytes, Erythro: erythrocytes.”

Additionally, we have amended the manuscript throughout to highlight that the Visium data is a prediction, and that when referring to Seurat-based spatial analyses, we refer to as predicted enrichment of the transcripts defining any given population, as below:

Line 725 “During infection, the transcripts associated with these subsets, as well as the mast cells and cDC1s, were predicted to be enriched in the subcutaneous adipose tissue (**Figure 3C**), mirroring the predicted localisation of other stromal cells driving immune cell recruitment and activation (**Figure 2I**). Furthermore, *Cd207⁺* LCs were predicted to be predominantly enriched in the epidermis/dermis of naïve animals but were also found within the adipose tissue in skin sections from infected animals, coinciding with adipocytes and several populations of progenitor keratinocyte populations (**Figure 1E, 3C, and S2B Figure**). The spatial enrichment of LCs in skin biopsies from naïve and infected animals was independently validated using smFISH, which confirms the accumulation of *Cd207⁺* LCs and *Cd4⁺* T cells around the perifollicular spaces in the epidermis/dermis in naïve samples, but were also detected in the adipose tissue within the hypodermis during infection (**Figure 3D**), consistent with previous studies demonstrating that CD207⁺ LCs display migratory behaviour during infection³⁷.

Line 797 “Spatial module scoring analysis predicted that *V γ 6⁺* cells localise mainly to the dermis and epidermis of naïve mice (**Figure 4E**), but their localisation changes in response to infection, where they are found to be enriched in the subcutaneous adipose tissue (**Figure 4E**).”

Reviewer #2 (Remarks to the Author):

The authors have significantly improved the manuscript. The four major aspects addressed in the point-by-point response letter, including FISH validation of the in silico predictions, have made the experimental evidence and the whole story much stronger. I have no more requests to make and would like to congratulate the authors for this important contribution, which goes beyond the parasitology and host-parasite interactions field.

Authors: We thank this reviewer for their positive assessment of our work, and for their time in providing feedback. We believe our manuscript is much improved now.

Reviewer #3 (Remarks to the Author):

The authors have thoroughly addressed all comments raised and provided additional experimental data to support the conclusions drawn.

Authors: We thank this reviewer for taking their time to review our manuscript.

I only have two minor suggestions for the authors' consideration, purely to help the readers with interpretation of the figures:

-Fig 2C: the order of the cell types with the colour code is discontinuous (IPA1 and 2),

Authors: We did this on purpose as we wanted to highlight these populations together within a box, as shown in Figure 2C (left panel), which links to the bar charts representing the estimated proportions of IPA1 and IPA2 on Figure 2C right panel. However, the colour scheme throughout figure 2A, 2B, and 2C) is consistent for the various cell types reported in the single cell dataset. We believe this makes the figure interpretation more straightforward.

-Fig 4I: label panels as naïve and infected as in Fig. 3D and 4F

Authors: We have now included labels on Figure 4I, as suggested.

REVIEWERS' COMMENTS

Reviewer #1 (Remarks to the Author):

The authors' are commended to evaluate spot-level cellular heterogeneity by carrying out cell type deconvolution analysis. While the analysis does not provide a satisfactory estimate of the cell type distribution at the spot level, it highlights the importance and difficulty of cell-type annotation which is hidden from the simpler best-matching-cell-type approach. Considering the main outcome of this study is 'a spatially resolved single cell atlas of the murine skin during T. brucei infection.' it is appropriate to explicitly acknowledge such limitation to avoid prevent over-interpretation.

Reviewer #3 (Remarks to the Author):

The authors have addressed all questions and concerns.

Decision on Nature Communications manuscript NCOMMS-23-09302C

We thank all the reviewers for their third round of comments. We hope that these additional clarifications satisfy their remaining concerns. Below we have provided point-by-point answers to the remaining comments raised by reviewer 1.

Reviewer #1 (Remarks to the Author):

The authors' are commended to evaluate spot-level cellular heterogeneity by carrying out cell type deconvolution analysis. While the analysis does not provide a satisfactory estimate of the cell type distribution at the spot level, it highlights the importance and difficulty of cell-type annotation which is hidden from the simpler best-matching-cell-type approach. Considering the main outcome of this study is 'a spatially resolved single cell atlas of the murine skin during *T. brucei* infection.' it is appropriate to explicitly acknowledge such limitation to avoid prevent over-interpretation.

Authors: We thank this reviewer for commending our efforts to provide the most accurate depiction possible of the data generated with our combined single cell and spatial transcriptomics dataset. We believe that the extend of the validation studies we have included to test the *in silico* predictions generated from our combined spatial approach supports the validity of the analysis presented in our study. As suggested by this reviewer, we have included the following additional text in the discussion to acknowledge the current limitations in the field:

Line 531: "It is important to acknowledge that although robust, the current strategies to deconvolve cell type distribution at the spot level for spatial transcriptomics (for example, *via* anchor-based integration approaches such as Seurat, or other models such as those provided by Giotto) rely heavily on the diversity of the single cell atlas used as an input. In our study, we offset this limitation by performing a series of additional validations, including *in situ* hybridisation and flow cytometry, of cell types and tissue regions of interest *in vivo*. We anticipate that as the field of spatial biology moves forward, additional packages and spot deconvolution strategies will be refined to capture additional nuances of the dataset."

Reviewer #3 (Remarks to the Author):

The authors have addressed all questions and concerns.

Authors: We once again thank this reviewer for taking the time to provide thorough and fair feedback on our manuscript. We believe the current version is much improved.